# DIFFUSION RANDOM FEATURE MODEL

## ABSTRACT

Diffusion probabilistic models have been successfully used to generate data from noise. However, most diffusion models are computationally expensive and difficult to interpret with a lack of theoretical justification. On the other hand, random feature models have gained popularity due to their interpretability but their application to complex machine learning tasks remains limited. In this work, we present a diffusion model-inspired deep random feature model that is interpretable and gives comparable numerical results to a fully connected neural network having the same number of trainable parameters. Specifically, we extend existing results for random features and derive generalization bounds between the distribution of sampled data and the true distribution using properties of score matching. We validate our findings by generating samples on the fashion MNIST (fMNIST) dataset and instrumental audio data.

## 1 INTRODUCTION

Generative modeling has been successfully used to generate a wide variety of data. Some well-known models are Generative Adversarial Networks (Goodfellow et al., 2014; Dinh et al., 2017), autoregressive models (Menick & Kalchbrenner, 2019), and variational autoencoders (Kingma & Welling, 2014). Remarkable results can also be obtained using energy-based modeling and score matching (Song & Ermon, 2020; Jolicoeur-Martineau et al., 2021; Jolicoeur-Martineau et al., 2021). Diffusion models are one such class of generative models that give exemplary performance in terms of data generation. A diffusion probabilistic model (or "diffusion model") is a parameterized model that is trained using variational inference to generate samples matching the data from input distribution after a finite number of timesteps. The model learns to reverse a diffusion process, which is a fixed Markov chain adding noise to the input data until it is destroyed. If the (Gaussian) noise added is very small, then the transition kernels are also conditional Gaussian distribution which leads to a neural network parameterization of the mean and the variance of the transition kernels. Most of the existing diffusion models are extremely complex and very computationally expensive. In this paper, we propose a model architecture to bridge the gap between interpretable models and diffusion models. Semi-random features, introduced by Kawaguchi et al. (2018) have been shown to have approximation powers beyond random features while preserving interpretability. Our main idea is to build a deep random feature model inspired by semi-random features and diffusion models. While there are numerous ways to define a training objective, we use the objective function derived in Ho et al. (2020) owing to its simplicity and ease of understanding.

Although there is a lot of work done in the field of both random features Rahimi & Recht (2007; 2008a;b); Kawaguchi et al. (2018); Saha et al. (2023b); Rudi & Rosasco (2017) and diffusion models Ho et al. (2020); Sohl-Dickstein et al. (2015); Song & Ermon (2019); Yang et al. (2022); Chen et al. (2023) we would like to highlight the significance of the proposed work and the key differences between the proposed and prior works.

- While the architecture of Kawaguchi et al. (2018) is similar to the proposed architecture, we would like to highlight the fact that our model architecture is capable of incorporating time as an input with a separate set of weights associated with it whereas in Kawaguchi et al. (2018) the authors only consider the input $\mathbf{x}$.

- Although our final result is based on results from Chen et al. (2023), we believe that the quantification of $\varepsilon_{\text{score}}$ in Chen et al. (2023) is a significant contribution to understanding the number of parameters needed (with respect to the input dimension and number of timesteps)

to obtain a small error. Our results show that an appropriate choice of model parameters can improve the bounds in Chen et al. (2023) from $TV(s(\mathbf{x}(T)), q(\mathbf{x}(0))) \leq \mathcal{O}(\varepsilon + \varepsilon_{\text{score}})$ to $TV(s(\mathbf{x}(T)), q(\mathbf{x}(0))) \leq \mathcal{O}(\varepsilon)$. .

Our paper is organized as follows. We give a brief background of diffusion models and random features in Section 2 along with the related works. The model architecture along with the algorithm and its theoretical results are given in Section 3. All experimental results are summarized in Section 4 followed by Section 5 where we conclude our paper and discuss future directions.

## 1.1 CONTRIBUTIONS

We propose a diffusion model-inspired random feature model, that uses semi-random-like features to learn the reverse process in the diffusion models. Our main contributions are given below:

- Our proposed model is the first of its kind to combine the idea of random features with generative models. It acts as a bridge between the theoretical and generative aspects of the diffusion model by providing approximation bounds of samples generated by diffusion model-based training algorithms. We show that for each fixed timestep, our architecture can be reduced to a random feature model preserving the properties of interpretability.

- Our numerical results on the fMNIST and the audio data validate our findings by generating samples from noise, as well as denoising signals not present in the training data. We show that even with a very small training set, our proposed model can denoise data and generate samples similar to training data.

## 2 BACKGROUND AND RELATED WORKS

We first recall some useful notations and terminologies corresponding to diffusion models. A $d-$ dimensional input sample is denoted by $\mathbf{x}_0$ and its (unknown) probability density function (p.d.f.) by $q(\mathbf{x}_0)$. We denote by $\mathbf{x}_k$, a vector corrupted with noise level $\beta_k$, where timesteps $k = 1, \ldots, K$ are discrete. The corresponding p.d.f. of the marginal distribution is denoted by $q(\mathbf{x}_k)$. With a large number of timesteps, the discretely corrupted samples can be written as a function of time denoted by $\mathbf{x}(t)$ for a continuous time variable $t \in [0, T]$ (for some terminal time $T$). The marginal distribution of $\mathbf{x}(t)$ is given by $q(\mathbf{x}(t))$ and $\mathcal{N}(\mathbf{0}, \mathbf{I})$ denotes the $d-$dimensional Gaussian distribution with the zero mean vector and the identity covariance matrix. We say that a function $p(\mathbf{x}) = \mathcal{N}(\boldsymbol{\mu}, \boldsymbol{\Sigma})$ to mean that $p(\mathbf{x})$ is the p.d.f. of a random vector $\mathbf{x}$ following the multivariate normal distribution with mean vector $\boldsymbol{\mu}$ and covariance matrix $\boldsymbol{\Sigma}$. Since our proposed method is motivated by the semi-random feature method proposed in Kawaguchi et al. (2018), a brief discussion is provided in Appendix A.1.

A diffusion model consists of two Markov chains: a forward process and a reverse process. The goal of the forward process is to degrade the input sample by gradually adding noise to it over a fixed number of timesteps. The reverse process involves learning to undo the added-noise steps using a parameterized model. Knowledge of the reverse process helps to generate new data starting with a random noisy vector followed by sampling through the reverse Markov chain (Koller & Friedman, 2009; Yang et al., 2022).

## 2.1 FORWARD PROCESS

Let $\mathbf{x}_0 \in \mathbb{R}^d$ be an input from an unknown distribution with p.d.f. $q(\mathbf{x}_0)$. Given a variance schedule $0 < \beta_1 < \beta_2 < ... < \beta_K < 1$, the forward process to obtain a degraded sample for a given timestep is defined as:

$$\mathbf{x}_{k+1} = \sqrt{1 - \beta_{k+1}}\mathbf{x}_k + \sqrt{\beta_{k+1}}\boldsymbol{\epsilon}_k, \quad \text{where } \boldsymbol{\epsilon}_k \sim \mathcal{N}(\mathbf{0}, \mathbf{I}) \text{ and } k = 0, \ldots, K-1. \quad (1)$$

The forward process generates a sequence of random variables $\mathbf{x}_1, \mathbf{x}_2, ..., \mathbf{x}_K$ with conditional distributions $q(\mathbf{x}_{k+1}|\mathbf{x}_k)$:

$$q(\mathbf{x}_{k+1}|\mathbf{x}_k) = \mathcal{N}(\mathbf{x}_{k+1}; \sqrt{1 - \beta_{k+1}}\mathbf{x}_k, \beta_{k+1}\mathbf{I}). \quad (2)$$

Let $\alpha_k = 1 - \beta_k$ for $k = 1, \ldots, K$ and $\overline{\alpha}_k = \prod_{i=1}^{k} \alpha_i$. Using the reparameterization trick, we can obtain $\mathbf{x}_k$ at any given time $k \in \{1, \ldots, K\}$ from $\mathbf{x}_0$:

$$\mathbf{x}_k = \sqrt{\alpha_k}\, \mathbf{x}_{k-1} + \sqrt{1 - \alpha_k}\, \boldsymbol{\epsilon}_{k-1} = \cdots = \sqrt{\overline{\alpha}_k}\, \mathbf{x}_0 + \sqrt{1 - \overline{\alpha}_k}\, \widetilde{\boldsymbol{\epsilon}}_0, \tag{3}$$

where $\widetilde{\boldsymbol{\epsilon}}_i \sim \mathcal{N}(\mathbf{0}, \mathbf{I})$ for $i = 0, \ldots, k-2$. Therefore, the conditional distribution $q(\mathbf{x}_{k+1}|\mathbf{x}_0)$ is

$$q(\mathbf{x}_{k+1}|\mathbf{x}_0) = \mathcal{N}(\mathbf{x}_{k+1}; \sqrt{\overline{\alpha}_{k+1}}\, \mathbf{x}_0, (1 - \overline{\alpha}_{k+1})\mathbf{I}), \tag{4}$$

At $k = K$, we have

$$\mathbf{x}_K = \sqrt{\overline{\alpha}_K}\, \mathbf{x}_0 + \sqrt{1 - \overline{\alpha}_K}\, \boldsymbol{\epsilon},$$

where $\boldsymbol{\epsilon} \sim \mathcal{N}(\mathbf{0}, \mathbf{I})$. Since $0 < \beta_1 < \ldots < \beta_K < 1$, $0 < \overline{\alpha}_K < \alpha_1^K < 1$. Using triangle inequality, we have $\lim_{K \to \infty} \overline{\alpha}_K = 0$. Therefore, $q(\mathbf{x}_K) \approx \mathcal{N}(\mathbf{0}, \mathbf{I})$, i.e., as the number of timesteps becomes very large, the distribution $q(\mathbf{x}_K)$ will approach the Gaussian distribution with mean $\mathbf{0}$ and covariance $\mathbf{I}$.

## 2.2 REVERSE PROCESS

The forward process degrades the input data such that $q(\mathbf{x}_K) \approx \mathcal{N}(\mathbf{0}, \mathbf{I})$. Our goal is to generate data from the input distribution by sampling from $q(\mathbf{x}_K)$ and gradually denoising, for which one needs to know the reverse distribution $q(\mathbf{x}_{k-1}|\mathbf{x}_k)$. In general, computation of $q(\mathbf{x}_{k-1}|\mathbf{x}_k)$ is intractable without the knowledge of $\mathbf{x}_0$ and hence we condition the reverse distribution on $\mathbf{x}_0$ in order to obtain the mean and variance for the reverse process. More precisely, using Bayes' rule, we have:

$$q(\mathbf{x}_{k-1}|\mathbf{x}_k, \mathbf{x}_0) = q(\mathbf{x}_k|\mathbf{x}_{k-1}, \mathbf{x}_0) \frac{q(\mathbf{x}_{k-1}, \mathbf{x}_0)}{q(\mathbf{x}_k, \mathbf{x}_0)} \frac{q(\mathbf{x}_0)}{q(\mathbf{x}_0)} = q(\mathbf{x}_k|\mathbf{x}_{k-1}, \mathbf{x}_0) \frac{q(\mathbf{x}_{k-1}|\mathbf{x}_0)}{q(\mathbf{x}_k|\mathbf{x}_0)}. \tag{5}$$

Plugging Equations (2), (3), and (4) to Equation (5), we conclude that $q(\mathbf{x}_{k-1}|\mathbf{x}_k, \mathbf{x}_0) = \mathcal{N}(\tilde{\boldsymbol{\mu}}_t, \tilde{\beta}_t)$ where

$$\tilde{\boldsymbol{\mu}}_k = \frac{1}{\sqrt{\alpha_k}} \left( \mathbf{x}_k - \frac{1 - \alpha_k}{\sqrt{1 - \bar{\alpha}_k}} \boldsymbol{\epsilon}_k \right) \text{ and } \tilde{\beta}_t = \frac{1 - \bar{\alpha}_{k-1}}{1 - \bar{\alpha}_k} \beta_k. \tag{6}$$

Our aim is to learn the reverse distribution from the obtained conditional reverse distribution. From Markovian theory, if $\beta_k$'s are small, the reverse process is also Gaussian (Sohl-Dickstein et al., 2015). Suppose $p_\theta(\mathbf{x}_{k-1}|\mathbf{x}_k)$ be the learned reverse distribution, then Markovian theory tells us that $p_\theta(\mathbf{x}_{k-1}|\mathbf{x}_k) = \mathcal{N}(\boldsymbol{\mu}_\theta(\mathbf{x}_k, k), \boldsymbol{\Sigma}_\theta(\mathbf{x}_k, k))$, where $\boldsymbol{\mu}_\theta(\mathbf{x}_k, k)$ and $\boldsymbol{\Sigma}_\theta(\mathbf{x}_k, k)$ are the learned mean vector and variance matrix respectively. Since the derived covariance matrix $\tilde{\beta}_k \mathbf{I}$ for conditional reverse distribution is constant, $\boldsymbol{\Sigma}_\theta(\mathbf{x}_k, k)$ need not be learnt. In Ho et al. (2020), the authors show that choosing $\boldsymbol{\Sigma}_\theta(\mathbf{x}_k, t)$ as $\beta_k \mathbf{I}$ or $\tilde{\beta}_k \mathbf{I}$ yield similar results and thus we fix $\boldsymbol{\Sigma}_\theta(\mathbf{x}_k, k) = \beta_k \mathbf{I}$ for simplicity. Furthermore, since $\mathbf{x}_k$ is also available as input to the model, the loss function derived in Ho et al. (2020) as a KL divergence between $q(\mathbf{x}_{k-1}|\mathbf{x}_k, \mathbf{x}_0)$ and $p_\theta(\mathbf{x}_{k-1}|\mathbf{x}_k)$ can be simplified as

$$D_{KL}(q(\mathbf{x}_{k-1}|\mathbf{x}_k, \mathbf{x}_0) \| p_\theta(\mathbf{x}_{k-1}|\mathbf{x}_k) = \mathbb{E}_q \left[ \frac{1}{2\beta_k^2} \| \tilde{\boldsymbol{\mu}}_k(\mathbf{x}_k, \mathbf{x}_0) - \boldsymbol{\mu}_\theta(\mathbf{x}_k, k) \|^2 \right] + \text{const} \tag{7}$$

Referring to the derivation in Ho et al. (2020) (refer to Appendix A.2 for more details), we can simplify Eq. (7) as:

$$\mathbb{E}_{k, \mathbf{x}_0, \boldsymbol{\epsilon}} \left[ \frac{1}{2\alpha_k(1 - \bar{\alpha}_k)} \| \boldsymbol{\epsilon} - \boldsymbol{\epsilon}_\theta(\mathbf{x}_k, k) \|^2 \right]. \tag{8}$$

where $\boldsymbol{\epsilon}_\theta$ now denotes a function approximator intended to predict the noise from $\mathbf{x}_k$. The above results show that we can either train the reverse process mean function approximator $\boldsymbol{\mu}_\theta$ to predict $\tilde{\boldsymbol{\mu}}_k$ or modify using its parameterization to predict $\boldsymbol{\epsilon}$. In our proposed algorithm, we choose to use the loss function from Eq. (8) since it is one of the simplest forms to train and understand. This formulation of DDPM also helps us to harness the power of SDEs in diffusion models through its connection to DSMs (Block et al., 2020)

## 2.3 DIFFUSION MODELS AND SDES

The forward process can also be generalized to stochastic differential equations (SDEs) if infinite time steps or noise levels are considered (SDEs) as proposed in Yang et al. (2022). To formulate the forward process as an SDE, let $t = \frac{k}{K}$ and define functions $\mathbf{x}(t), \beta(t)$ and $\boldsymbol{\epsilon}(t)$ such that $\mathbf{x}(\frac{k}{K}) = \mathbf{x}_k$, $\beta(\frac{k}{K}) = K\beta_k$ and $\boldsymbol{\epsilon}(\frac{k}{K}) = \boldsymbol{\epsilon}_k$. Note that in the limit $K \to \infty$, we get $t \in [0, 1]$. Using the derivations from Yang et al. (2022), the forward process can be written as a SDE of the form,

$$d\mathbf{x} = \frac{-\beta(t)}{2}\mathbf{x}dt + \sqrt{\beta(t)}d\mathbf{w} \tag{9}$$

where $\mathbf{w}$ is the standard Wiener process. The above equation now is in the form of an SDE

$$d\mathbf{x} = f(\mathbf{x}, t)dt + g(t)d\mathbf{w} \tag{10}$$

where $f(\mathbf{x}, t)$ and $g(t)$ are diffusion and drift functions of the SDE respectively, and $\mathbf{w}$ is a standard Wiener process. The above processed can be reversed by solving the reverse-SDE,

$$d\mathbf{x} = [f(\mathbf{x}, t) - g(t)^2 \nabla_{\mathbf{x}(t)} \log q(\mathbf{x}(t))]dt + g(t)d\bar{\mathbf{w}} \tag{11}$$

where $\bar{\mathbf{w}}$ is a standard Wiener process backwards in time, and $dt$ denotes an infinitesimal negative time step and $q(\mathbf{x}(t))$ is the marginal distribution of $\mathbf{x}(t)$. Note that in particular for Eq.(9), the reverse SDE will be of the form

$$d\mathbf{x} = \left[\frac{\beta(t)}{2} - \beta(t)\nabla_{\mathbf{x}(t)} \log q(\mathbf{x}(t))\right] dt + \sqrt{\beta(t)}d\bar{\mathbf{w}} \tag{12}$$

The unknown term $\nabla_{\mathbf{x}(t)} \log q(\mathbf{x}(t))$ is called the score function and is estimated by training a parameterized model $s_\theta(\mathbf{x}(t), t)$ via minimization of the loss given by

$$\mathbb{E}_{q(\mathbf{x}(t))} \left[\frac{1}{2}\left\|s_\theta(\mathbf{x}(t), t) - \nabla_{\mathbf{x}(t)} \log q(\mathbf{x}(t))\right\|_2^2\right] \tag{13}$$

Note that since $q(\mathbf{x}(0))$ is unknown, therefore the distribution $q(\mathbf{x}(t))$ and subsequently the score function $\nabla_{\mathbf{x}(t)} \log q(\mathbf{x}(t))$ are also unknown. Referring to results from Vincent (2011), we see that the loss in Eq. (13) is equivalent to the denoising score matching (DSM) objective given by,

$$\mathbb{E}_{q(\mathbf{x}(t), \mathbf{x}(0))} \left[\frac{1}{2}\left\|s_\theta(\mathbf{x}(t), t) - \nabla_{\mathbf{x}(t)} \log q(\mathbf{x}(t)|\mathbf{x}(0))\right\|_2^2\right]. \tag{14}$$

We can see that the above objective is the same as the objective of DSM in the discrete setting. Apart from the success of score based models using SDEs, an additional advantage of formulating the diffusion model using SDEs is the theoretical analysis based on results from SDEs. In our paper, we aim to use this connection to build a theoretical understanding of our proposed model.

While there are remarkable results for improving training and sampling for diffusion models, little has been explored in terms of the model architectures. Since distribution learning and data generation is a complex task, it is unsurprising that conventional diffusion models are computationally expensive. From previous works in Ho et al. (2020), U-Net (or variations on U-Net combined with ResNet, CNNs, etc.) architecture is still the most commonly used model for diffusion models. The architecture of U-Net is designed in a way that it preserves the input dimension of the data i.e., the output and input dimension are the same. Additionally, U-Nets generally consist of a contracting (downsampling) and expanding (upsampling) path. The contracting path consists of sequences of convolutional and max-pooling layers that downsample the input images and extracts the layers. However, all these architectures have more than one hundred million parameters making training (and sampling) cumbersome. An alternative approach to reduce the complexity of machine learning algorithms is to use a random feature model (RFM) (Rahimi & Recht, 2007; 2008a) for approximating the kernels using a randomized basis. RFMs are derived from kernel-based methods which utilize a pre-defined nonlinear function basis called kernel $K(\mathbf{x}, \mathbf{y})$. From the neural network point of view, an RFM is a two-layer network with a fixed single hidden layer sampled randomly (Rahimi & Recht, 2007; 2008a). Not only do random feature-based methods give similar results to that of a shallow network, but the model in itself is also interpretable which makes it a favorable method to use. Some recent works

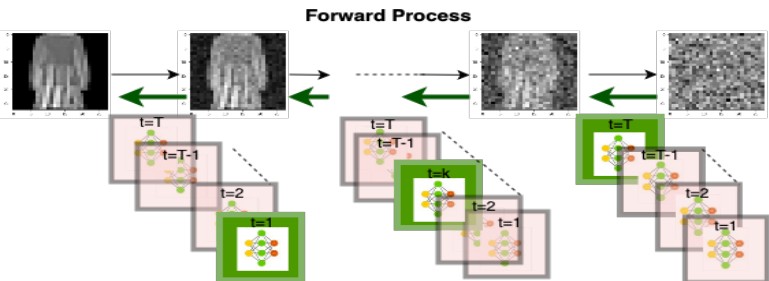

Figure 1: Representation of DRFM. The green boxes denote the random feature layer which is active corresponding to the timestep selected, while the other remain fixed.

which use random features for a variety of tasks are explored in Saha et al. (2023b;a); Choromanski et al. (2021; 2022); Richardson et al. (2023); Nelsen & Stuart (2021). However, random features can lack expressibility due to their structure and thus we aim to propose an architecture that can be more flexible in learning yet retaining properties of random features. Inspired by semi-random features (Kawaguchi et al., 2018), DDPM (Ho et al., 2020) and the idea of building deep random features, our diffusion random feature model serves as a potential alternative to the existing diffusion model architectures.

## 3 ALGORITHM AND THEORY

Our proposed model is a diffusion model inspired random feature model. The main idea of our model is to build an interpretable deep random feature architecture for diffusion models. Our work is inspired by *denoising diffusion probabilistic model* (DDPM) proposed in Ho et al. (2020) and semi-random features proposed in Kawaguchi et al. (2018). Let $\mathbf{x}_0 \in \mathbb{R}^d$ be the input data belonging to an unknown distribution $q(\mathbf{x}_0)$. Let $K$ denote the total number of timesteps in which the forward process is applied. Suppose $N$ is the number of features. For each timestep $k$, we build a noise predictor function $p_\theta$ of the form

$$p_\theta(\mathbf{x}_k, k) = (\sin(\mathbf{x}_k^T \mathbf{W} + \mathbf{b}^T) \odot \cos(\boldsymbol{\tau}_k^T \boldsymbol{\theta}^{(1)}))\boldsymbol{\theta}^{(2)}, \tag{15}$$

where $\mathbf{x}_k \in \mathbb{R}^d$, $\mathbf{W} \in \mathbb{R}^{d \times N}$, $\mathbf{b} = [b_1 \quad \dots \quad b_N]^T \in \mathbb{R}^N$, $\boldsymbol{\theta}^{(1)} = (\theta_{ki}^{(1)}) \in \mathbb{R}^{K \times N}$, $\boldsymbol{\tau}_k \in \mathbb{R}^K$, $\boldsymbol{\theta}^{(2)} = (\theta_{ij}^{(2)}) \in \mathbb{R}^{N \times d}$, and $\odot$ denotes element-wise multiplication. The vector $\boldsymbol{\tau}_k$ $(k \geq 1)$ is a one-hot vector with the position of one corresponding to the timestep $k$. The motivation to use trainable weights corresponding to the time parameter is twofold: first, we want to associate importance to the timestep being used when optimizing the weights; secondly, we aim to build a deep random feature model layered through time. The inspiration for using cosine as an activation comes from the idea of positional encoding used for similar tasks. In general, positional encoding remains fixed, but for our method, we wish to make the weights associated with timestep random and trainable. This is done so that the model learns the noise level associated with the timestep. Our aim is to train the parameters $\boldsymbol{\theta} = \{\boldsymbol{\theta}^{(1)}, \boldsymbol{\theta}^{(2)}\}$ while $\mathbf{W}$ and $\mathbf{b}$ are randomly sampled and fixed. The model is trained using Algorithm 1.

### 3.1 THEORETICAL RESULTS

We provide theoretical results corresponding to our proposed model. We first formulate our proposed model as a time-dependent layered random features model, followed by the proof of obtaining sample generation bounds. The obtained bounds help to prove that DRFM is capable of generating samples from the distribution on which it was trained using the results from Chen et al. (2023).

---

**Algorithm 1** Training and sampling using DRFM

---

**Require:** Sample $\mathbf{x}_0 \sim q(\mathbf{x}_0)$ where $q$ is an unknown distribution, variance schedule $\beta = \{\beta_1, ..., \beta_K\}$ such that $0 < \beta_1 < \beta_2 < ... < \beta_K < 1$, random weight matrix $\mathbf{W} = [w_{ij}]$ and bias vector $\mathbf{b}$ sampled from a distribution $\rho$ and total number of epochs `epoch`.
**Ensure:**

Training

1: Choose random timestep $k \in \{1, 2, ..., K\}$ and build vector $\boldsymbol{\tau}_k = [0, ...0, 1, 0, ..., 0]^T$ where 1 is in $k^{th}$ position.
2: Define the forward process for $k = 1, 2, ..., K$ as

$$\mathbf{x}_k = \sqrt{1 - \beta_k}\mathbf{x}_{k-1} + \sqrt{\beta_k}\boldsymbol{\epsilon}_k$$

where $\boldsymbol{\epsilon}_k \sim \mathcal{N}(\mathbf{0}, \mathbf{I})$.
3: **for** $j$ in `epochs` **do**
4:      $k \sim \mathcal{U}\{1, 2, ..., K\}$.
5:      Define $\boldsymbol{\tau}_k$ as in line 1.
6:      $p_\theta(\mathbf{x}_k, k) \leftarrow \left(\sin(\mathbf{x}_k^T\mathbf{W} + \mathbf{b}) \odot \cos(\boldsymbol{\tau}_k^T\boldsymbol{\theta^{(1)}})\right) \boldsymbol{\theta^{(2)}}$
7:      Update $\boldsymbol{\theta} = [\boldsymbol{\theta}^{(1)}, \boldsymbol{\theta}^{(2)}]$ by minimizing the loss $L = \dfrac{1}{K}\sum_{k=1}^{K} \left\|\boldsymbol{\epsilon}_k - p_\theta(\mathbf{x}_k, k)\right\|_2^2$.

8: **end for**

Sampling

9: Sample a point $\mathbf{x}_K \sim \mathcal{N}(\mathbf{0}, \mathbf{I})$
10: **for** $k = K - 1, ..., 1$ **do**
11:      Sample $\boldsymbol{\epsilon} \sim \mathcal{N}(\mathbf{0}, \mathbf{I})$

12:      $\tilde{\mathbf{x}}_{k-1} = \dfrac{1}{\sqrt{1 - \beta_k}}\left(\mathbf{x}_k - \dfrac{\sqrt{\beta_k}}{\sqrt{1 - \prod_{i=1}^{k}(1 - \beta_i)}}p_\theta(\mathbf{x}_k, k))\right) + \beta_k\boldsymbol{\epsilon}$

13: **end for**
     **Output:** Generated sample $\tilde{\mathbf{x}}_0$

---

For a fixed timestep $k$, Eq. 15 can be written as:

$$
\begin{aligned}
p_\theta(\mathbf{x}_k, k) &= (\sin(\mathbf{x}_k^T\mathbf{W} + \mathbf{b}^T) \odot \cos(\boldsymbol{\tau}_k\boldsymbol{\theta}^{(1)}))\boldsymbol{\theta}^{(2)} \\
&= \sin(\mathbf{x}_k^T\mathbf{W} + \mathbf{b}^T)
\begin{bmatrix}
\cos(\theta_{k1}^{(1)})\theta_{11}^{(2)} & ... & \cos(\theta_{k1}^{(1)})\theta_{1d}^{(2)} \\
\vdots & ... & \vdots \\
\cos(\theta_{kN}^{(1)})\theta_{N1}^{(2)} & ... & \cos(\theta_{kN}^{(1)})\theta_{Nd}^{(2)}
\end{bmatrix}.
\end{aligned}
\tag{16}
$$

Thus, for a fixed time $k$, our proposed architecture is a random feature model with a fixed dictionary having $N$ features denoted by $\phi(\langle \mathbf{x}_k, \boldsymbol{\omega}_i \rangle) = \sin(\mathbf{x}_k^T\boldsymbol{\omega}_i + b_i), \forall i = 1, \dots, N$ and learnt coefficients $\mathbf{C} = (c_{ij}) \in \mathbb{R}^{N \times d}$ whose entries are $c_{ij} = \cos(\boldsymbol{\theta}_{ki}^{(1)})\boldsymbol{\theta}_{ij}^{(2)}, \ \forall i = 1, \dots, N; j = 1, \dots, d$.

As depicted in Figure 1, DRFM can be visualized as $K$ random feature models stacked up in a column. Each random feature model has associated weights corresponding to its timestep which also gets optimized implicitly while training. The reformulation of DRFM into multiple random feature models leads us to our first Lemma stated below. We show that the class of functions generated by our proposed model is the same as the class of functions approximated by random feature models.

**Lemma 3.1.** *Let $\mathcal{G}_{k,\boldsymbol{\omega}}$ denote the set of functions that can be approximated by DRFM at each timestep $k$ defined as*

$$\mathcal{G}_{k,\boldsymbol{\omega}} = \left\{ \boldsymbol{g}(\mathbf{x}) = \sum_{j=1}^{N} \cos(\boldsymbol{\theta}_{kj}^{(1)}) \boldsymbol{\theta}_{j}^{(2)} \phi(\mathbf{x}_k^T \boldsymbol{\omega}_j) \Big| \left\|\boldsymbol{\theta}_j^{(2)}\right\|_{\infty} \leq \frac{C}{N} \right\} \tag{17}$$

*where $\boldsymbol{\theta}_j^{(2)}$ denotes the $j^{th}$ row of the matrix $\boldsymbol{\theta}^{(2)} \in \mathbb{R}^{N \times d}$. Then for a fixed $k$ and $\mathcal{F}_{\boldsymbol{\omega}}$ defined in Equation (18), $\mathcal{G}_{k,\boldsymbol{\omega}} = \mathcal{F}_{\boldsymbol{\omega}}$ where*

$$\mathcal{F}_{\boldsymbol{\omega}} = \left\{ \boldsymbol{f}(\mathbf{x}) = \sum_{j=1}^{N} \boldsymbol{\alpha_j}\, \phi(\mathbf{x}^T \boldsymbol{\omega}_j) \Big| \|\boldsymbol{\alpha_j}\|_{\infty} \leq \frac{C}{N} \right\}. \tag{18}$$

In the next lemma stated below, we extend results from Rahimi & Recht (2007; 2008b;a) to find approximation error bounds for vector-valued functions.

**Lemma 3.2.** *Let $X \subset \mathbb{R}^d$ denote the training dataset and suppose $q$ is a measure on X, and $\boldsymbol{f}^*$ a function in $\mathcal{F}_\rho$ where*

$$\mathcal{F}_\rho = \left\{ \boldsymbol{f}(\mathbf{x}) = \int_\Omega \boldsymbol{\alpha}(\boldsymbol{\omega})\phi(\mathbf{x};\boldsymbol{\omega})\, d\boldsymbol{\omega} \ \Big| \ \|\boldsymbol{\alpha}(\boldsymbol{\omega})\|_{\infty} \leq C\rho(\boldsymbol{\omega}) \right\}.$$

*If $[\boldsymbol{\omega}_j]_{j \in [N]}$ are drawn iid from $\rho$, then for $\delta > 0$, with probability at least $1 - \delta$ over $[\boldsymbol{\omega}_j]_{j \in [N]}$, there exists a function $\boldsymbol{f}^\sharp \in \mathcal{F}_{\boldsymbol{\omega}}$ such that*

$$\|\boldsymbol{f}^\sharp - \boldsymbol{f}^*\|_2 \leq \frac{C\sqrt{d}}{\sqrt{N}}\left(1 + \sqrt{2\log\frac{1}{\delta}}\right), \tag{19}$$

*where $\mathcal{F}_{\boldsymbol{\omega}}$ is defined above in Eq. (18).*

Using the above-stated Lemmas and results given in Chen et al. (2023), we derive our main theorem. Specifically, we quantify the total variation between the distribution learned by our model and the true data distribution.

**Theorem 3.3.** *For a given probability density $q(\mathbf{x}_0)$ on $\mathbb{R}^d$ suppose the following conditions hold:*

1. *For all $t \geq 0$, the score $\nabla \log q(\mathbf{x}(t))$ is $L-$Lipschitz.*

2. *For $\eta > 0$, $\mathbb{E}_{q(\mathbf{x}_0)}[\|.\|^{2+\eta}] < \infty$.*

*Let $p_\theta(\mathbf{x}_0, 0)$ be the sample generated by DRFM after $K$ timesteps at terminal time $T$. Then for the SDE formulation of the DDPM algorithm, if the step size $h := T/K$ satisfies $h \lesssim 1/L$, where $L \geq 1$. Then,*

$$TV(p_\theta(\mathbf{x}_0, 0), q(\mathbf{x}_0)) \lesssim \sqrt{KL(q(\mathbf{x}_0)\|\gamma)}\exp(-T) + (L\sqrt{dh} + Lm_2 h)\sqrt{T} + \frac{C_2\sqrt{TKd}}{\sqrt{N}}\left(1 + \sqrt{2\log\frac{1}{\delta}}\right) \tag{20}$$

*where $C_2 \geq \max\limits_{1 \leq i \leq N, 1 \leq j \leq d} \left|\boldsymbol{\theta}_{ij}^{(2)}\right|$ and $\gamma$ is the p.d.f. of the multivariate normal distribution with mean vector $\mathbf{0}$ and covariance matrix $\mathbf{I}$.*

The error bound given in Eq. (20) consists of three types of errors: (i) convergence error of the forward process; (ii) discretization error of the associated SDE with step size $h > 0$; and (iii) score estimation error, respectively. Note that the first two terms are independent of the model architecture. While for most models score estimation is difficult, our main contribution involves quantifying that error which gives us an estimate of the number of parameters are needed for the third error to become negligible. The proofs have been deferred till the appendix.

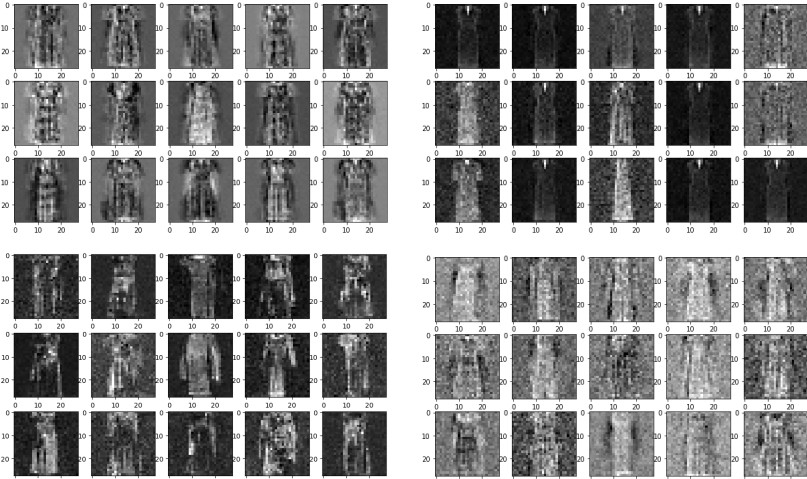

Figure 2: Figures generated from random noise when trained on 100 "dress" images with 100 timesteps linearly spaced between $10^{-4}$ and 0.02. The models were trained using 30000 epochs using ADAM optimizer with a learning rate $\eta = 0.001$ by minimizing the MSE loss. **Top left block:** DRFM. **Top right block:** NN. **Bottom left block:** U-Net. **Bottom right block:** RF.

## 4 EXPERIMENTAL RESULTS

In order to validate our findings, we train our proposed model on both image and audio data. We evaluate the validity of our model on two tasks: (i) generating data from noise and; (ii) denoising data corrupted with Gaussian noise. The experiments on images were done by taking one hundred images of the class "dress" and "shoes" separately from the fMNIST dataset. For audio data, we use two music samples corresponding to the flute and guitar. Results for the audio dataset are given in Appendix A.7.4. We compare our method with a fully connected version of DRFM (denoted by NN) where we train $[\mathbf{W}, \mathbf{b}, \boldsymbol{\theta}^{(1)}, \boldsymbol{\theta}^{(2)}]$ while preserving the number of trainable parameters, a U-Net model (details given in the Appendix) and a classical random feature approach with only $\boldsymbol{\theta}^{(2)}$ being trainable (denoted by RF). We create the image dataset for our experiments by considering 100 images of size $28 \times 28$ taken from a particular class of fMNIST dataset. DRFM is trained with 80000 random features. For NN, U-Net, and RF we adjust the number of trainable parameters to match that of DRFM. We use 100 equally spaced timesteps for the forward diffusion process between $10^{-4}$ and 0.02 and train for 30000 epochs by minimizing the mean squared error (MSE) loss using ADAM optimizer with a learning rate of $\eta = 0.001$. For the task of generating images, we generated fifteen samples from randomly sampled noise. In our second task we give fifteen images from the same class (but not in the training set) corrupted with noise and test if our model can denoise the image.

Results from Figure 2 demonstrate that our method learns the overall features of the input distribution. Although the model is trained with a small size of training data (only one hundred images) and timesteps (one hundred timesteps), we can see that the samples generated from pure noise have already started to resemble the input distribution of dresses. The samples generated by the U-Net model also simply learns to generate the overall features of the distribution. The overall sample quality is not significantly better than the ones generated by DRFM. For NN model we note that most of the generated samples are the same with a dark shadow while for RF model, the generated images are very noisy and barely recognizable.

We also test our models ability to remove noise from images. We take fifteen random images of "dress" not in the training data and corrupt it with $20\%$ noise. The proposed model is used for denoising. In Figure 3 we can see that the model can recover a denoised image which is in fact better than the results obtained when sampling from pure noise. The U-Net and NN models performs quite well for most of the images. However for few cases with the NN model, it fails to denoise anything and the final image remains noisy. RF model fails to denoise and the resultant images still contain noise. Table 1 gives the Fréchet Inception Distance (FID) calculated using a batch of 50 images from the training dataset and 15 of the generated images by each of the models. Note that the scores values

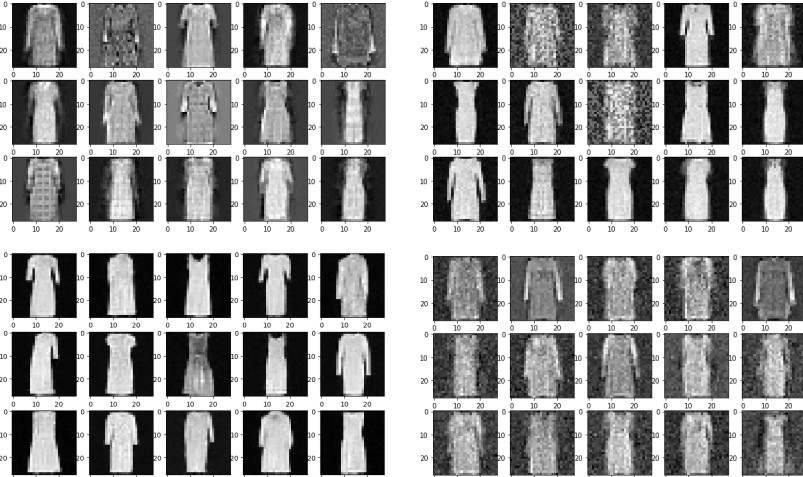

Figure 3: Figures denoised from images corrupted with 20 % noise. We use 100 timesteps linearly spaced between $10^{-4}$ and 0.02. The models were trained using 30000 epochs using ADAM optimizer with a learning rate $\eta = 0.001$ by minimizing the MSE loss. **Top left block:** DRFM. **Top right block:** NN. **Bottom left block:** U-Net. **Bottom right block:** RF.

| Model | Timesteps | FID Score |
|-------|-----------|-----------|
| DRFM  | 100       | **453.87** |
| U-Net | 100       | 463.28    |
| NN    | 100       | 457.21    |
| RF    | 100       | 470.12    |

| Model | Timesteps | FID Score |
|-------|-----------|-----------|
| DRFM  | 100       | 394.99    |
| U-Net | 100       | 388.38    |
| NN    | 100       | **378.18** |
| RF    | 100       | 450.23    |

Table 1: FID scores for all the models when trained with 100 equally spaced timesteps between $10^{-4}$ and 0.02. **Right:** FID scores for generative task. **Left:** FID scores for denoising task.

given are for the sake of comparing the four methods and more samples may improve the FID score. We see that for the generative task, the the proposed DRFM architecture gives the lowest scores. The more commonly used U-Net model is at the third position after the NN model. For the denoising task, we see that the NN gives the best results, (although some of the images are inconsistently noisy), followed by U-Net and then DRFM. Note that for U-Net, while most images are denoised perfectly, some images are incompletly formed which leads to a lower FID score. Moreover, we see that all the FID scores are also consistent with the qualitative assessment made in Figures 2 and 3. In order to check the effect of the number of timesteps on the sampling power of DRFM, we also run our model using 1000 timesteps between $10^{-4}$ and 0.02. The images generated/denoised are given in Appendix A.7. The model performance seem to improve significantly only for the generative task and not the denoising task, which is expected as more reverse steps would be required to generate a point in the input distribution.

## 5 CONCLUSION

In this paper, we proposed an interpretable diffusion random feature model with quantified upper bounds on the samples generated by DRFM with respect to the input distribution. We validated our findings using numerical experiments on audio and a subset of fMNIST dataset. Comparisons with a fully connected network (when all layers are trainable), a simple U-Net architecture, and random features method (all but the last layer is fixed i.e., only $\theta_2$ is trainable) highlighted the advantages of our model which performed better or comparably to a fully connected network and U-Net, and better than the random features method. However, the fully connected network and U-Net cannot be interpreted with quantifiable bounds for function approximation. Further direction of this work involves extending DRFM beyond its shallow nature into deeper architecture to avoid the curse of dimensionality with respect to the number of features required for approximation.

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
