# A APPENDIX

We provide a brief discussion on semi-random features from Kawaguchi et al. (2018) and some derivations from previous works on diffusion models (Song & Ermon, 2019; Sohl-Dickstein et al., 2015; Ho et al., 2020) for the sake of completeness for the readers. The proof of our main result and some additional experimental results are also provided. Appendix A.1 gives a brief discussion on semi-random features, Appendix A.2 gives details of deriving the mean vector and covariance matrix of the reverse distribution, and in Appendix A.3 we give a detailed proof of the equivalence between the loss functions of DDPM and DSM. We also provide proof of our proposed Lemma 3.2 and 3.1 in Appendices A.5 and A.4 respectively, followed by some additional experimental results in Appendix A.7.

## A.1 SEMI-RANDOM FEATURES (SRF)

We briefly discuss the semi-random features introduced in Kawaguchi et al. (2018). Given input $\mathbf{x} \in \mathbb{R}^d$, a shallow semi-random features approximation is defined as,

$$\hat{f}(\mathbf{x}) = \left( \sigma_s(\mathbf{x}^T \mathbf{W}) \odot \left( \mathbf{x}^T \boldsymbol{\theta}^{(1)} \right) \right) \boldsymbol{\theta}^{(2)} \tag{21}$$

where $\odot$ denotes the Hadamard product of two matrices, $N$ denotes the number of features, $\mathbf{W} \in \mathbb{R}^{d \times N}$ are the fixed parameters, $\boldsymbol{\theta}^{(1)} \in \mathbb{R}^{d \times N}$ and $\boldsymbol{\theta}^{(4)} \in \mathbb{R}^N$ are the trainable parameters, and $\sigma_s(\cdot)$ is a non-linear function defined by $\sigma_s(z) = z^s H(z)$ where $H$ is the Heaviside step function ($H(z) = 1$ for $z > 0$ and 0 otherwise). The authors show that despite the non-convex learning problem, the shallow SRF model has no bad local minimum. Since our DRFM model is proposed with a similar architecture which can be reformulated as a random feature model, the DRFM training also avoids a bad local minima.

## A.2 MEAN AND VARIANCE OF REVERSE DISTRIBUTIONS

We know from Markovian theory that if the variances $\beta_1, \ldots, \beta_k$ are small enough, then the reverse process $q(\mathbf{x}_{k-1}|\mathbf{x}_k)$ will also be Gaussian i.e., $q(\mathbf{x}_{k-1}|\mathbf{x}_k) = \mathcal{N}(\mathbf{x}_{k-1}; \tilde{\boldsymbol{\mu}}_t, \tilde{\beta}_t)$. Since the input distribution $q(\mathbf{x}_0)$ is unknown, $\tilde{\boldsymbol{\mu}}_t$ and $\tilde{\beta}_t$ are intractable unless conditioned on $\mathbf{x}_0$, in which case we can derive $\tilde{\boldsymbol{\mu}}_t$ and $\tilde{\beta}_t$ following previous works. For completeness, we provide a detailed derivation in the Lemma below.

**Lemma A.1.** *Let $q(\mathbf{x}_0)$ denote the probability density function of an unknown distribution and $0 < \beta_1 < \cdots < \beta_T < 1$ be a sequence of variances such that $q(\mathbf{x}_t|\mathbf{x}_{t-1}) = \mathcal{N}(\mathbf{x}_t; \sqrt{1-\beta_t}\mathbf{x}_{t-1}, \beta_t I_d)$. Then for the reverse Markov chain conditioned on $\mathbf{x}_0$, denoted by $q(\mathbf{x}_{t-1}|\mathbf{x}_t, \mathbf{x}_0)$, the mean vector and the covariance matrix is $\tilde{\boldsymbol{\mu}}_t = [\tilde{\mu}_t, \cdots, \tilde{\mu}_t]^T \in \mathbb{R}^d$ and $\tilde{\boldsymbol{\Sigma}}_t = \tilde{\beta}_t I_d \in \mathbb{R}^{d \times d}$ respectively such that,*

$$\tilde{\mu}_t = \frac{\sqrt{\alpha_t}(1-\bar{\alpha}_{t-1})}{1-\bar{\alpha}_t}\mathbf{x}_t + \frac{\sqrt{\bar{\alpha}_{t-1}}\beta_t}{1-\bar{\alpha}_t}\mathbf{x}_0 \text{ and } \tilde{\beta}_t = \frac{1-\bar{\alpha}_{t-1}}{1-\bar{\alpha}_t}\beta_t$$

*Proof.*

$$q(\mathbf{x}_{k-1}|\mathbf{x}_k, \mathbf{x}_0)$$

$$= q(\mathbf{x}_k|\mathbf{x}_{k-1}, \mathbf{x}_0) . \frac{q(\mathbf{x}_{k-1}, \mathbf{x}_0)}{q(\mathbf{x}_k, \mathbf{x}_0)} . \frac{q(\mathbf{x}_0)}{q(\mathbf{x}_0)} \quad \text{(Baye's rule)}$$

$$= q(\mathbf{x}_k|\mathbf{x}_{k-1}, \mathbf{x}_0) . \frac{q(\mathbf{x}_{k-1}|\mathbf{x}_0)}{q(\mathbf{x}_k|\mathbf{x}_0)}$$

$$= \frac{1}{\sqrt{(2\pi\beta_k)^d}} \exp\left(-\frac{1}{2}\frac{(\mathbf{x}_k - \sqrt{\alpha_k}\mathbf{x}_{k-1})^T(\mathbf{x}_k - \sqrt{\alpha_k}\mathbf{x}_{k-1})}{\beta_k}\right)$$

$$\cdot \frac{1}{\sqrt{(2\pi(1-\bar{\alpha}_{k-1}))^d}} \exp\left(-\frac{1}{2}\frac{(\mathbf{x}_{k-1} - \sqrt{\bar{\alpha}_{k-1}}\mathbf{x}_0)^T(\mathbf{x}_{k-1} - \sqrt{\bar{\alpha}_{k-1}}\mathbf{x}_0)}{1-\bar{\alpha}_{k-1}}\right)$$

$$\cdot \left(\sqrt{(2\pi(1-\bar{\alpha}_k))^d}\right) \exp\left(\frac{1}{2}\frac{(\mathbf{x}_k - \sqrt{\bar{\alpha}_k}\mathbf{x}_0)^T(\mathbf{x}_k - \sqrt{\bar{\alpha}_k}\mathbf{x}_0)}{1-\bar{\alpha}_k}\right) \tag{22}$$

$$= \frac{\sqrt{(1-\bar{\alpha}_k)^d}}{\sqrt{(2\pi\beta_k(1-\bar{\alpha}_{k-1}))^d}} \exp\left(-\frac{1}{2}\left[\frac{\mathbf{x}_k^T\mathbf{x}_k - \sqrt{\alpha_k}\mathbf{x}_k^T\mathbf{x}_{k-1} - \sqrt{\alpha_k}\mathbf{x}_{k-1}^T\mathbf{x}_k + \alpha_k\mathbf{x}_{k-1}^T\mathbf{x}_{k-1}}{\beta_k}\right]\right)$$

$$\cdot \exp\left(-\frac{1}{2}\left[\frac{\mathbf{x}_{k-1}^T\mathbf{x}_{k-1} - \sqrt{\bar{\alpha}_{k-1}}\mathbf{x}_0^T\mathbf{x}_{k-1} - \sqrt{\bar{\alpha}_{k-1}}\mathbf{x}_{k-1}^T\mathbf{x}_0 + \bar{\alpha}_{k-1}\mathbf{x}_0^T\mathbf{x}_0}{1-\bar{\alpha}_{k-1}}\right]\right)$$

$$\cdot \exp\left(\frac{1}{2}\left[\frac{\mathbf{x}_k^T\mathbf{x}_k - \sqrt{\bar{\alpha}_k}\mathbf{x}_k^T\mathbf{x}_0 - \sqrt{\bar{\alpha}_k}\mathbf{x}_0^T\mathbf{x}_k + \bar{\alpha}_k\mathbf{x}_0^T\mathbf{x}_0}{1-\bar{\alpha}_k}\right]\right) \quad (23)$$

Simplifying Eq.(23), the terms inside the exponential function becomes,

$$\mathbf{x}_{k-1}^T\mathbf{x}_{k-1}\left(\frac{\alpha_k}{\beta_k} + \frac{1}{1-\bar{\alpha}_{k-1}}\right) - \mathbf{x}_{k-1}^T\left(\frac{\sqrt{\alpha_k}}{\beta_k}\mathbf{x}_k + \frac{\sqrt{\bar{\alpha}_{k-1}}}{1-\bar{\alpha}_{k-1}}\mathbf{x}_0\right)$$

$$- \mathbf{x}_{k-1}\left(\frac{\sqrt{\alpha_k}}{\beta_k}\mathbf{x}_k^T + \frac{\sqrt{\bar{\alpha}_{k-1}}}{1-\bar{\alpha}_{k-1}}\mathbf{x}_0^T\right) + \mathbf{x}_k^T\mathbf{x}_k\left(\frac{1}{\beta_k} - \frac{1}{1-\bar{\alpha}_k}\right) + \frac{\sqrt{\bar{\alpha}_k}}{1-\bar{\alpha}_k}\mathbf{x}_k^T\mathbf{x}_0$$

$$+ \frac{\sqrt{\bar{\alpha}_k}}{1-\bar{\alpha}_k}\mathbf{x}_0^T\mathbf{x}_k + \mathbf{x}_0^T\mathbf{x}_0\left(\frac{\bar{\alpha}_{k-1}}{1-\bar{\alpha}_{k-1}} - \frac{\bar{\alpha}_k}{1-\bar{\alpha}_k}\right) \quad (24)$$

$$= \mathbf{x}_{k-1}^T\mathbf{x}_{k-1}\left(\frac{\alpha_k}{\beta_k} + \frac{1}{1-\bar{\alpha}_{k-1}}\right) - \mathbf{x}_{k-1}^T\left(\frac{\sqrt{\alpha_k}}{\beta_k}\mathbf{x}_k + \frac{\sqrt{\bar{\alpha}_{k-1}}}{1-\bar{\alpha}_{k-1}}\mathbf{x}_0\right) - \mathbf{x}_{k-1}\left(\frac{\sqrt{\alpha_k}}{\beta_k}\mathbf{x}_k^T + \frac{\sqrt{\bar{\alpha}_{k-1}}}{1-\bar{\alpha}_{k-1}}\mathbf{x}_0^T\right)$$

$$+ \left(\mathbf{x}_k\sqrt{\frac{1}{\beta_k} - \frac{1}{1-\bar{\alpha}_k}} + \mathbf{x}_0\sqrt{\frac{\bar{\alpha}_{k-1}}{1-\bar{\alpha}_{k-1}} - \frac{\bar{\alpha}_k}{1-\bar{\alpha}_{k-1}}}\right)^T\left(\mathbf{x}_k\sqrt{\frac{1}{\beta_k} - \frac{1}{1-\bar{\alpha}_k}} + \mathbf{x}_0\sqrt{\frac{\bar{\alpha}_{k-1}}{1-\bar{\alpha}_{k-1}} - \frac{\bar{\alpha}_k}{1-\bar{\alpha}_{k-1}}}\right)$$

$$(25)$$

$$= \left(\mathbf{x}_{k-1}\sqrt{\frac{\alpha_k}{\beta_k} + \frac{1}{1-\bar{\alpha}_{k-1}}} - \left(\mathbf{x}_k\sqrt{\frac{1}{\beta_k} - \frac{1}{1-\bar{\alpha}_k}} + \mathbf{x}_0\sqrt{\frac{\bar{\alpha}_{k-1}}{1-\bar{\alpha}_{k-1}} - \frac{\bar{\alpha}_k}{1-\bar{\alpha}_{k-1}}}\right)\right)^T$$

$$\cdot \left(\mathbf{x}_{k-1}\sqrt{\frac{\alpha_k}{\beta_k} + \frac{1}{1-\bar{\alpha}_{k-1}}} - \left(\mathbf{x}_k\sqrt{\frac{1}{\beta_k} - \frac{1}{1-\bar{\alpha}_k}} + \mathbf{x}_0\sqrt{\frac{\bar{\alpha}_{k-1}}{1-\bar{\alpha}_{k-1}} - \frac{\bar{\alpha}_k}{1-\bar{\alpha}_{k-1}}}\right)\right) \quad (26)$$

$$= \left(\frac{\alpha_k}{\beta_k} + \frac{1}{1-\bar{\alpha}_{k-1}}\right)\left(\mathbf{x}_{k-1} - \left(\mathbf{x}_k\frac{\sqrt{\frac{1}{\beta_k} - \frac{1}{1-\bar{\alpha}_k}}}{\sqrt{\frac{\alpha_k}{\beta_k} + \frac{1}{1-\bar{\alpha}_{k-1}}}} + \mathbf{x}_0\frac{\sqrt{\frac{\bar{\alpha}_{k-1}}{1-\bar{\alpha}_{k-1}} - \frac{\bar{\alpha}_k}{1-\bar{\alpha}_{k-1}}}}{\sqrt{\frac{\alpha_k}{\beta_k} + \frac{1}{1-\bar{\alpha}_{k-1}}}}\right)\right)^T$$

$$\cdot \left(\mathbf{x}_{k-1} - \left(\mathbf{x}_k\frac{\sqrt{\frac{1}{\beta_k} - \frac{1}{1-\bar{\alpha}_k}}}{\sqrt{\frac{\alpha_k}{\beta_k} + \frac{1}{1-\bar{\alpha}_{k-1}}}} + \mathbf{x}_0\frac{\sqrt{\frac{\bar{\alpha}_{k-1}}{1-\bar{\alpha}_{k-1}} - \frac{\bar{\alpha}_k}{1-\bar{\alpha}_{k-1}}}}{\sqrt{\frac{\alpha_k}{\beta_k} + \frac{1}{1-\bar{\alpha}_{k-1}}}}\right)\right) \quad (27)$$

Thus combining Equations (23) and (27) we get,

$$q(\mathbf{x}_{k-1}|\mathbf{x}_k, \mathbf{x}_0) = \frac{\sqrt{(1-\bar{\alpha}_k)^d}}{\sqrt{(2\pi\beta_k(1-\bar{\alpha}_{k-1}))^d}} \exp\left[-\frac{1}{2}\left(\frac{\alpha_k}{\beta_k} + \frac{1}{1-\bar{\alpha}_{k-1}}\right)\right]$$

$$\left(\mathbf{x}_{k-1} - \left(\mathbf{x}_k\frac{\sqrt{\frac{1}{\beta_k} - \frac{1}{1-\bar{\alpha}_k}}}{\sqrt{\frac{\alpha_k}{\beta_k} + \frac{1}{1-\bar{\alpha}_{k-1}}}} + \mathbf{x}_0\frac{\sqrt{\frac{\bar{\alpha}_{k-1}}{1-\bar{\alpha}_{k-1}} - \frac{\bar{\alpha}_k}{1-\bar{\alpha}_{k-1}}}}{\sqrt{\frac{\alpha_k}{\beta_k} + \frac{1}{1-\bar{\alpha}_{k-1}}}}\right)\right)^T$$

$$\left[\left(\mathbf{x}_{k-1} - \left(\mathbf{x}_k\frac{\sqrt{\frac{1}{\beta_k} - \frac{1}{1-\bar{\alpha}_k}}}{\sqrt{\frac{\alpha_k}{\beta_k} + \frac{1}{1-\bar{\alpha}_{k-1}}}} + \mathbf{x}_0\frac{\sqrt{\frac{\bar{\alpha}_{k-1}}{1-\bar{\alpha}_{k-1}} - \frac{\bar{\alpha}_k}{1-\bar{\alpha}_{k-1}}}}{\sqrt{\frac{\alpha_k}{\beta_k} + \frac{1}{1-\bar{\alpha}_{k-1}}}}\right)\right)\right].$$

Thus we see that the probability density function of the reverse distribution conditioned on $\mathbf{x}_0$ is also a Gaussian distribution with mean vector $\tilde{\boldsymbol{\mu}}_t$ and covariance matrix $\tilde{\beta}_t \mathbf{I}$ which are given by,

$$\tilde{\boldsymbol{\mu}}_t = \frac{\sqrt{\alpha_t}(1 - \bar{\alpha}_{t-1})}{1 - \bar{\alpha}_t}\mathbf{x}_t + \frac{\sqrt{\bar{\alpha}_{t-1}}\beta_t}{1 - \bar{\alpha}_t}\mathbf{x}_0 \text{ and } \tilde{\beta}_t = \frac{1 - \bar{\alpha}_{k-1}}{1 - \bar{\alpha}_k}\beta_k \tag{28}$$

since, $\mathbf{x}_0 = \frac{1}{\sqrt{\bar{\alpha}}}(\mathbf{x}_k - \sqrt{1 - \bar{\alpha}_k}\boldsymbol{\epsilon}_k)$. $\qquad\square$

### A.3 DDPM AND DSM

We can also apply the DDPM algorithm for score matching by formulating the DDPM objective as a DSM objective.

$$L_{\text{DDPM}} = \mathbb{E}_{k,\mathbf{x}_0,\boldsymbol{\epsilon}} \left[ \frac{1}{2\alpha_k(1 - \bar{\alpha}_k)} \|\boldsymbol{\epsilon} - \boldsymbol{\epsilon}_\theta(\mathbf{x}_k, k)\|^2 \right] \tag{29}$$

$$= \mathbb{E}_{k,\mathbf{x}_0,\boldsymbol{\epsilon}} \left[ \frac{1}{2\alpha_k(1 - \bar{\alpha}_k)} \left\| \frac{\mathbf{x}_k - \sqrt{\bar{\alpha}_k}\mathbf{x}_0}{\sqrt{1 - \bar{\alpha}_k}} - \boldsymbol{\epsilon}_\theta(\mathbf{x}_k, k) \right\|^2 \right] \tag{30}$$

$$= \mathbb{E}_{k,\mathbf{x}_0,\mathbf{x}_k} \left[ \frac{1}{2\alpha_k(1 - \bar{\alpha}_k)} \left\| \frac{\mathbf{x}_k - \sqrt{\bar{\alpha}_k}\mathbf{x}_0}{1 - \bar{\alpha}_k}\sqrt{1 - \bar{\alpha}_k} - \frac{\sqrt{1 - \bar{\alpha}_k}}{\sqrt{1 - \bar{\alpha}_k}}\boldsymbol{\epsilon}_\theta(\mathbf{x}_k, k) \right\|^2 \right] \tag{31}$$

$$= \mathbb{E}_{k,\mathbf{x}_0,\mathbf{x}_k} \left[ \frac{1}{2\alpha_k} \left\| -\nabla_{\mathbf{x}_k} \log q(\mathbf{x}_k|\mathbf{x}_0) - \frac{1}{\sqrt{1 - \bar{\alpha}_k}}\boldsymbol{\epsilon}_\theta(\mathbf{x}_k, k) \right\|^2 \right] \tag{32}$$

$$= \mathbb{E}_{k,\mathbf{x}_0,\mathbf{x}_k} \left[ \frac{1}{2\alpha_k} \left\| s_\theta(\mathbf{x}_k, k) - \nabla_{\mathbf{x}_k} \log q(\mathbf{x}_k|\mathbf{x}_0) \right\|^2 \right] = L_{\text{DSM}} \tag{33}$$

where $s_\theta(\mathbf{x}_k, k) = -\frac{1}{\sqrt{1 - \bar{\alpha}_k}}\boldsymbol{\epsilon}_\theta(\mathbf{x}_k, k)$. The above formulation is known as denoising score matching (DSM), which is equivalent to the objective of DDPM. Furthermore, the objective of DSM is also related to the objective of score based generative models using SDEs (Song & Ermon, 2019). We briefly discuss the connection between diffusion models, SDEs and DSM in the upcoming section (Song & Ermon, 2019; Yang et al., 2022; Ho et al., 2020).

### A.4 PROOF OF LEMMA 3.1

*Proof.* The above equality can be proved easily.

Fix $k$. Consider $\boldsymbol{g}(\mathbf{x}) \in \mathcal{G}_{k,\boldsymbol{\omega}}$, then $\boldsymbol{g}(\mathbf{x}) = \sum_{j=1}^{N} \cos(\boldsymbol{\theta}_{kj}^{(1)})\boldsymbol{\theta}_j^{(2)}\phi(\mathbf{x}_k^T\boldsymbol{\omega}_j)$. Clearly, $\boldsymbol{g}(\mathbf{x}) \in \mathcal{F}_{\boldsymbol{\omega}}$ as $\left\| \cos(\boldsymbol{\theta}_{kj}^{(1)})\boldsymbol{\theta}_j^{(2)} \right\|_\infty \leq \left\| \boldsymbol{\theta}_j^{(2)} \right\|_\infty \leq \frac{C}{N}$. Thus $\mathcal{G}_{k,\boldsymbol{\omega}} \subseteq \mathcal{F}_{\boldsymbol{\omega}}$.

Conversely let $\boldsymbol{f}(\mathbf{x}) \in \mathcal{F}_{\boldsymbol{\omega}}$, then $\boldsymbol{f}(\mathbf{x}) = \sum_{j=1}^{N} \boldsymbol{\alpha}_j \phi(\mathbf{x}_k^T\boldsymbol{\omega}_j)$. Choose $\boldsymbol{\theta}_j^{(2)} = \boldsymbol{\alpha}_j$ and $\boldsymbol{\theta}_{kj}^{(1)} = [0, \cdots, 0]$.

As $\cos(\boldsymbol{\theta}_{kj}^{(1)})\boldsymbol{\theta}_j^{(2)} = \boldsymbol{\alpha}_j$, thus $\boldsymbol{f}(\mathbf{x}) = \sum_{j=1}^{N} \cos(\boldsymbol{\theta}_{kj}^{(1)})\boldsymbol{\theta}_j^{(2)}\phi(\mathbf{x}^T\boldsymbol{\omega}_j)$ and $\|\boldsymbol{\theta}_j^{(2)}\|_\infty = \|\boldsymbol{\alpha}_j\|_\infty \leq \frac{C}{N}$. Hence $\boldsymbol{f}(\mathbf{x}) \in \mathcal{G}(\mathbf{x}_k, k)$ and $\mathcal{F}_{\boldsymbol{\omega}} \subseteq \mathcal{G}_{k,\boldsymbol{\omega}}$. $\qquad\square$

### A.5 PROOF OF LEMMA 3.2

*Proof.* We follow the proof technique described in Rahimi & Recht (2008b). As $\boldsymbol{f}^* \in \mathcal{F}_\rho$, then $\boldsymbol{f}^*(\mathbf{x}) = \int_{\boldsymbol{\Omega}} \boldsymbol{\alpha}(\boldsymbol{\omega})\phi(\mathbf{x}; \boldsymbol{\omega})d\boldsymbol{\omega}$. Construct $\boldsymbol{f}_k = \boldsymbol{\beta}_k\phi(\cdot; \boldsymbol{\omega}_k)$, $k = 1, \cdots, N$ such that

$$\boldsymbol{\beta}_k = \frac{\boldsymbol{\alpha}(\boldsymbol{\omega}_k)}{\rho(\boldsymbol{\omega}_k)} = \frac{1}{\rho(\boldsymbol{\omega}_k)} \begin{bmatrix} \alpha_1(\boldsymbol{\omega}_k) \\ \vdots \\ \alpha_d(\boldsymbol{\omega}_k) \end{bmatrix}.$$

Note that $\mathbb{E}_{\boldsymbol{\omega}}(\boldsymbol{f}_k) = \int_{\boldsymbol{\omega}} \frac{\boldsymbol{\alpha}(\boldsymbol{\omega}_k)}{\rho(\boldsymbol{\omega}_k)} \phi(\mathbf{x}; \boldsymbol{\omega}) \rho(\boldsymbol{\omega}_k) d\boldsymbol{\omega} = \boldsymbol{f}^*.$

Define the sample average of these functions as $\boldsymbol{f}^\sharp(\mathbf{x}) = \sum_{k=1}^{N} \frac{\boldsymbol{\beta}_k}{N} \phi(\mathbf{x}; \boldsymbol{\omega}_k).$

As $\left\| \frac{\boldsymbol{\beta}_k}{N} \right\|_\infty \le \frac{C}{N}$, thus $\boldsymbol{f}^\sharp \in \mathcal{F}_{\boldsymbol{\omega}}$. Also note that $\|\boldsymbol{\beta}_k \phi(\cdot; \boldsymbol{\omega}_k)\|_2 \le \sqrt{d} \|\|\boldsymbol{\beta}_k \phi(\cdot; \boldsymbol{\omega}_k)\|_\infty \le \sqrt{d} C.$

In order to getthe desired result, we use McDiarmid's inequality. Define a scaler function on $F = \{\boldsymbol{f}_1, \cdots, \boldsymbol{f}_N\}$ as $g(F) = \|\boldsymbol{f}^\sharp - \mathbb{E}_F \boldsymbol{f}^\sharp\|_2$. We claim that the function $g$ is stable under perturbation of its $i$th argument.

Define $\tilde{F} = \{\boldsymbol{f}_1, \cdots, \tilde{\boldsymbol{f}}_i, \cdots, \boldsymbol{f}_N\}$ i.e., $\tilde{F}$ differs from $F$ only in its $i$th element. Then

$$\left| g(F) - g(\tilde{F}) \right| = \left| \left\| \boldsymbol{f}^\sharp - \mathbb{E}_F \boldsymbol{f}^\sharp \right\|_2 - \left\| \tilde{\boldsymbol{f}}^\sharp - \mathbb{E}_{\tilde{F}} \boldsymbol{f}^\sharp \right\|_2 \right| \le \left\| \boldsymbol{f}^\sharp - \tilde{\boldsymbol{f}}^\sharp \right\|_2 \tag{34}$$

where the above inequality is obtained from triangle inequality. Further,

$$\left\| \boldsymbol{f}^\sharp - \tilde{\boldsymbol{f}}^\sharp \right\|_2 = \frac{1}{N} \left\| \boldsymbol{f}_i - \tilde{\boldsymbol{f}}_i \right\|_2 \le \left\| (\boldsymbol{\beta}_i - \tilde{\boldsymbol{\beta}}_i) \phi(\cdot; \boldsymbol{\omega}) \right\|_2 \le \frac{\sqrt{d} C}{N}. \tag{35}$$

Thus $\mathbb{E}[g(F)^2] = \mathbb{E}\left[ \left\| \boldsymbol{f}^\sharp - \mathbb{E}_F \boldsymbol{f}^\sharp \right\|_2^2 \right] = \frac{1}{N} \left[ \mathbb{E}\left[ \left\| \sum_{k=1}^{N} \boldsymbol{f}_k \right\|_2^2 \right] - \left\| \mathbb{E}\left[ \sum_{k=1}^{N} \boldsymbol{f}_k \right] \right\|_2^2 \right].$

Since $\|\boldsymbol{f}_k\|_2 \le \sqrt{d} C$, using Jensen's inequality and above result we get

$$\mathbb{E}[g(F)] \le \sqrt{\mathbb{E}(g^2(F))} \le \frac{\sqrt{d} C}{\sqrt{N}}. \tag{36}$$

Finally the required bounds can be obtained by combining above result and McDiarmid's inequality. $\square$

## A.6 Proof of Theorem 3.3

**Lemma A.2** (from Chen et al. (2023)). *For a given data distribution $q(\mathbf{x}(0)$ on $\mathbb{R}^d$, terminal time $T$ and step size $h = T/K$, suppose the following assumptions hold.*

1. *For $t \ge 0$, the score $\nabla \log q(\mathbf{x}(t))$ is $L-$Lipschitz.*

2. *For $\eta > 0$, $\mathbb{E}_{q(\mathbf{x}_0)}[\|\cdot\|^{2+\eta}] < \infty$.*

3. *The error in score estimate is bounded in $L^2$,*

$$E_{q_{\mathbf{x}(t)}}[\|s(\mathbf{x}(t)) - \nabla \ln q(\mathbf{x}(t))\|^2] \le E^2.$$

*If the step size satisfies $h \lesssim 1/L$, where $L \ge 1$. Then,*

$$TV(s(\mathbf{x}(T))), q(\mathbf{x}_0)) \lesssim \sqrt{KL(q(\mathbf{x}_0)\|\gamma)} \exp(-T) + (L\sqrt{dh} + Lm_2 h)\sqrt{T} + E\sqrt{T} \tag{37}$$

*where, $\gamma$ is the p.d.f. of the multivariate normal distribution with mean vector $\mathbf{0}$ and covariance matrix $\mathbf{I}$.*

We can combine the proof of Lemma 3.1, 3.2, and A.2 to get the required bounds.

## A.7 Additional results

### A.7.1 U-Net Architecture

We use a simple U-Net architecture with the same number of trainable layers as that of DRFM. The representation is given below. Note that a positional time encoding layer is added before each downsampling step.

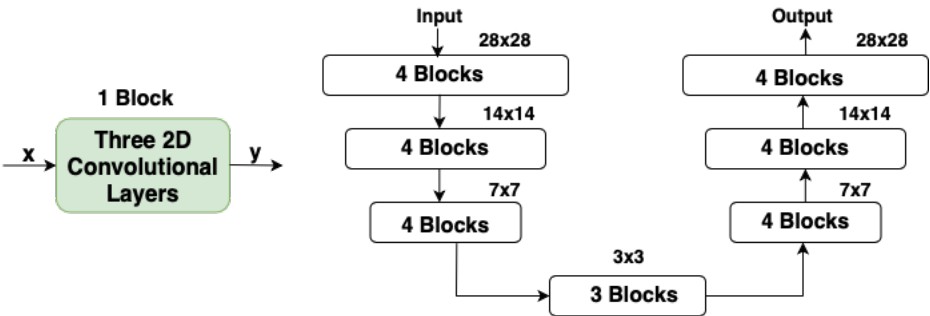

Figure 4: U-Net architecture used in the paper. **Left:** One block consisting if three convolutional layers. **Right:** U-Net architecture with downsampling and upsampling layers.

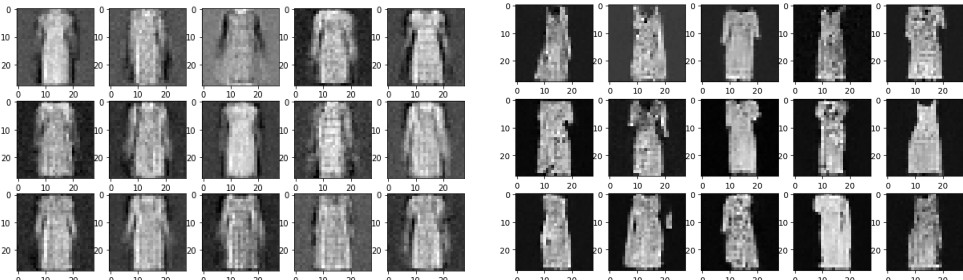

Figure 5: Generated images trained on 100 "dress" images with 1000 timesteps. **Top left block:** DRFM. **Top right block:** U-Net.

### A.7.2 EXPERIMENT WITH 1000 TIMESTEPS

In order to check the effect of the number of timesteps on the sampling power of DRFM, we also run our model using 1000 timesteps between $10^{-4}$ and 0.02. The images generated are given in Figure 5. Samples generated from noise seem to improve with the increase in the number of timesteps for both DRFM and U-Net. The improvement in sample quality with an increased number of timesteps for generating data is expected as more reverse steps would be required to generate a point in the input distribution than for the task of denoising.

### A.7.3 EXPERIMENTS ON A DIFFERENT CLASS OF DATA

We also conducted experiments with a different class of data with the same setup as discussed in the main text. This time we selected the class "shoes" and tested our model's performance. The conclusions drawn support the claims we made with the previous class of data where DRFM can denoise images well while only learning to generate basic features of the model class when generating from pure noise. The plots for the above are depicted in Figure 6.

### A.7.4 RESULTS ON AUDIO DATA

Our second experiment involves learning to generate and denoise audio signals. We use one data sample each taken from two different instruments, namely guitar and flute. There are a total of 5560 points for each instrument piece. We train our model using 15000 random features with 100 timesteps taken between $10^{-4}$ and 0.02 for 30000 epochs. The samples are generated from pure noise using the trained model to remove noise for each reverse diffusion step. We also test if our model is capable of denoising a signal when it is not a part of the training set explicitly (but similar to it). For that we use a validation data point containing samples from a music piece when both guitar and flute are played simultaneously. We plot the samples generated from pure Gaussian noise in Figure 7.

The plots in Figure 7 demonstrate the potential of DRFM to generate signals not a part of the original training data. Figure 7(b) shows that there is no advantage of using a NN model since the results are

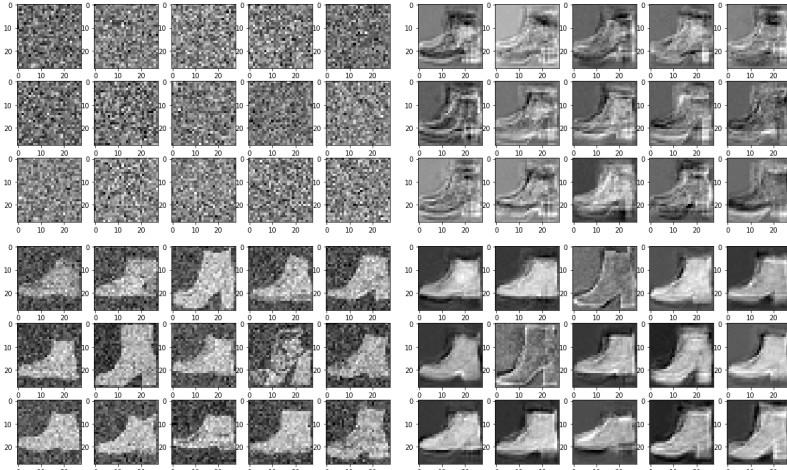

Figure 6: Samples generated from noise and noisy images when trained on 100 images of "shoes". **Top left block:** Gaussian noise. **Top right block:** Generated samples. **Bottom left block:** Noisy input. **Bottom right block:** Denoised images.

much worse. The network does not learn anything and signal generated is just another noise. On the other hand, our proposed model DRFM generates signals which are similar to the original two signals used to train the data.

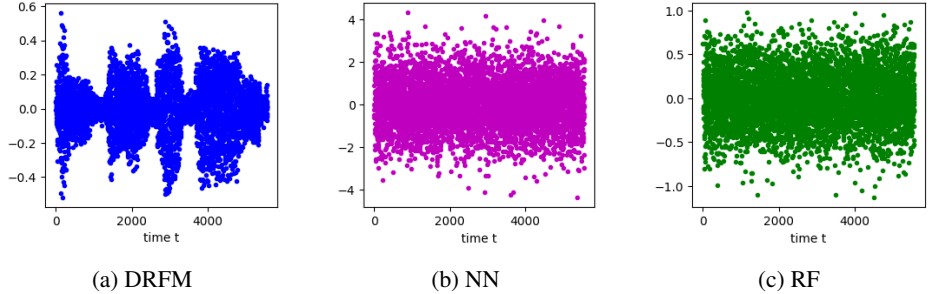

(a) DRFM              (b) NN              (c) RF

Figure 7: Generated samples using different models.

Figure 8 shows that when our trained model is applied to a validation data point, it can successfully recover and denoise the signal. This is more evident when the signal is played as an audio file. There are however some extra elements which get added while recovering due to the presence of noise which is a common effect of using diffusion models for denoising.

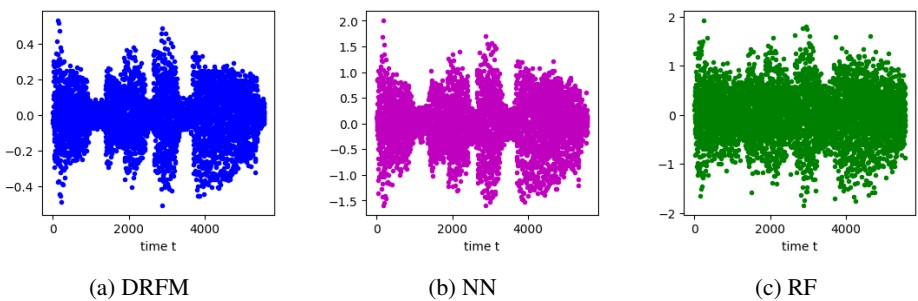

(a) DRFM              (b) NN              (c) RF

Figure 8: Denoised signals using different models.