# OpenReview forum: "Diffusion Random Feature Model"
_ICLR.cc/2024/Conference — ICLR 2024 Conference Withdrawn Submission_

### Official Review · Reviewer_crxL · 2023-10-30

**Soundness:** 3 good
**Presentation:** 2 fair
**Contribution:** 2 fair
**Rating:** 5
**Confidence:** 4

**Summary:**

This paper presents the Diffusion Random Feature Models (DRFMs) as a solution to ongoing challenges in both interpretability and computational complexity of diffusion models. The introduced DRFM yields numerical outcomes comparable to those of a fully connected neural network with an equivalent number of trainable parameters. From a theoretical perspective, the authors expand the random features and establish generalization bounds between the sampled data distribution and the actual distribution, leveraging score-matching properties. The efficacy of the proposed method is validated experimentally using samples from the Fashion MNIST dataset and instrumental audio data.

**Strengths:**

- The presentation is overall clear and easy to follow.

- The overall method is clearly stated. The mathematical notations are well-defined, but some are redundant. The theoretical results are sound. The authors have made comprehensive proof and have quantified the error in the estimation, which is good.

- The authors expand the random features and establish generalization bounds between the sampled data distribution and the actual distribution, leveraging score-matching properties, making contributions in both perspectives.

**Weaknesses:**

### Novelty

- The proposed framework is inspired by boththe denoising diffusion probabilistic model introduced by Ho et al. (2020), and the semi-random features introduced by Kawaguchi et al. (2018). It seems the proposed Diffusion Random Feature Model (DRFM) is incremental in integrating the above two works. Specifically, in Kawaguchi et al. (2018), a parameterization of one single-hidden layer is defined in a similar form to Eq (15). The authors employ this parameterization for the noise-prediction network in training DDPM.

- Throughout the paper, it is not straightforward to understand the advantage of the parameterization in Eq (15) in terms of interpretability and computation complexity. Any analysis or experimental results should be provided to address these challenges stated in the introduction.

### Technical quality
- The experiments validate the efficacy of the proposed DRFM. However, when considering the challenges diffusion models face in terms of interpretability and computation, the experiments seem to omit direct comparisons. Additionally, many of the presented results are not quantified using specific metrics, as observed in Fig 2. This makes it challenging to identify the model's superior performance.

- The paper currently compares the DRFM solely with a fully-connected layer. Given the prominence of both the Unet and transformer architectures in noise prediction networks, it would be beneficial for the authors to discuss the relationship of DRFM to these models. In particular, considering that transformer layers utilize a combination of the attention mechanism and fully-connected MLPs, they align more closely with the scope of the fully-connected layer baseline. It would be valuable to explore this connection further.

- Since the authors have generalized the random features with the properties of score-matching, it remains unknown whether DRFM can improve the performance in the tasks evaluated in Kawaguchi et al. (2018).

### Presentation quality
- In the abstract, the authors state that diffusion models suffer from a lack of theoretical justification. This assertion is somewhat unclear and could benefit from further specificity. It would be helpful if the authors could elucidate whether this deficiency refers to aspects within statistical, optimization, or other theoretical domains.

- Mathematical notations are redundant and inconsistent. For example,
$\tilde \epsilon_i$ does not seem to be used anywhere in section 2.1 the following sections.

- Since this work is inspired by the DDPM, and the semi-random features. The authors need to introduce the semi-random features liked done in section 2. Without the knowledge of semi-random features, it is hard to understand the importance and contribution of the proposed method.

**Questions:**

Please see the point-to-point review in the Weakness section.

---

> ### Author Response · Authors · 2023-11-22
> **Response to comments on novelty**
>
> *It seems the proposed Diffusion Random Feature Model (DRFM) is incremental in integrating the above two works. Specifically, in Kawaguchi et al. (2018), a parameterization of one single-hidden layer is defined in a similar form to Eq (15). The authors employ this parameterization for the noise-prediction network in training DDPM.*
>
> **Answer:**
> -  We would like to highlight the fact that our model architecture is capable of incorporating time as an input with a separate set of weights associated with it. This is much different from the architecture proposed in Kawaguchi et al, (2018) where the authors only consider the input $\mathbf{x}$.
>
> - We would like to point out that although our final result is based on results from Chen et. al. (2022), we believe that the quantification of $\varepsilon_{\text{score}}$ is a significant contribution to understand the number of parameters needed (with respect to the input dimension and number of timesteps) to obtain a small error. This quantification helps us to choose the appropriate number of model parameters before training. In particular, note that for appropriate choices of $T$ and $h$, the bounds in Chen et. al. (2022), can be atmost reduced to $TV(p_{\theta}(0),q)\leq\mathcal{O}(\varepsilon+\varepsilon_{\text{score}})$. Our results show that we can also choose the parameters of the model so that $TV(p_{\theta}(0),q)\leq\mathcal{O}(\varepsilon)$.
>
> *Throughout the paper, it is not straightforward to understand the advantage of the parameterization in Eq (15) in terms of interpretability and computation complexity. Any analysis or experimental results should be provided to address these challenges stated in the introduction.*
>
> **Answer:**
>
> - The advantage of the proposed architecture lies in being able to write it as a random feature model which helps us to obtain the results given in the paper and quantifying the the error bounds. The bounds show that by choosing $N\gg Kd$ for our model, we can reduce the bound $TV(p_{\theta}(0),q)\leq\mathcal{O}(\varepsilon+\varepsilon_{\text{score}})$ to $TV(p_{\theta}(0),q)\leq\mathcal{O}(\varepsilon)$. We included a brief description of the signicance of our work in the revised version of the paper.
>
> - Our choice of the model in (15) is based on Fourier-type feature functions as they represent the Gaussian kernel when the components of matrix $\mathbf{W}$ are sampled from Gaussian distribution and bias vector $\mathbf{b}$ is sampled from the uniform distribution. Since the aim of diffusion models is to learn the mean (and covariance) of the reverse process whose transition kernels are Gaussian distributions, we choose this particular form. For more details on the relation between Fourier-type features and Gaussian kernel, we refer to Rahimi and Recht (2007).

---

> ### Author Response · Authors · 2023-11-22
> **Response to comments on presentation quality**
>
> *1. In the abstract, the authors state that diffusion models suffer from a lack of theoretical justification. This assertion is somewhat unclear and could benefit from further specificity. It would be helpful if the authors could elucidate whether this deficiency refers to aspects within statistical, optimization, or other theoretical domains.*
>
> **Answer:**
>
> We mention that diffusion models suffer from a lack of theoretical justification to indicate that two aspects:
> - Parameterized models are a black box: The models used to train diffusion models are too complex w.r.t. the number of layers, architecture, etc and can not be analyzed for generalization bounds.
> - Optimization process: The number of results analyzing the training/sampling process is limited with some significant results based on SDE formulation of DDPM given in Chen et. al. (2022) , Lee et. al. (2022).
> For our proposed work, we focus on the first aspect as there are already existing results based on the optimization aspect.
>
> *2. Mathematical notations are redundant and inconsistent. For example,
> $\tilde{\epsilon}_i$ does not seem to be used anywhere in section 2.1 the following sections.*
>
> **Answer:**
>
> When we use the reparameterization trick, we use the notation $\tilde{\epsilon}_i$ to differentiate it from $\epsilon_i$. Note that although $\tilde{\epsilon}_i$ is also a Gaussian distribution, it is obtained by taking a sum of Gaussian distributions of the previous and current timestep. Although in practice both $\epsilon$ and $\tilde{\epsilon}_i$ are sampled from $\mathcal{N}(0,I)$, they are derived differently and hence the separate notation.
>
> *3. Since this work is inspired by the DDPM, and the semi-random features. The authors need to introduce the semi-random features liked done in section 2. Without the knowledge of semi-random features, it is hard to understand the importance and contribution of the proposed method.*
>
> **Answer:**
>
> Although our method is inspired by the semi-random features, the proposed architecture is different in terms of formulation and applications. In Kawaguchi et al, (2018) the authors only consider the input
> $\mathbf{x}$. However, as per the suggestion we have added a brief description of semi-random features in the Appendix for completeness and reference purposes (see revised supplementary material).

---

> ### Author Response · Authors · 2023-11-22
>
> Thank you for reviewing our manuscript. We appreciate the constructive feedback. We have carefully updated the manuscript to address your concerns. We have marked the changes in the main text in blue.

---

> ### Author Response · Authors · 2023-11-22
> **Responses to comments on technical quality**
>
> *1. The experiments validate the efficacy of the proposed DRFM. However, when considering the challenges diffusion models face in terms of interpretability and computation, the experiments seem to omit direct comparisons. Additionally, many of the presented results are not quantified using specific metrics, as observed in Fig 2. This makes it challenging to identify the model's superior performance.*
>
> **Answer:**
>
> - Table 1 (see revised version of the paper and also given below) gives the Fr\'echet Inception Distance (FID) calculated using a batch of 50 images from the training dataset and 15 of the generated images by each of the models. Note that the scores values given are for the sake of comparing the four methods and more samples may improve the FID score. We see that for the generative task, the proposed DRFM architecture gives the lowest scores. The more commonly used U-Net model is at the third position after the NN model. For the denoising task, we see that the NN gives the best results, (although some of the images are inconsistently noisy), followed by U-Net and then DRFM. Note that for U-Net, while most images are denoised perfectly, some images are incompletely formed which leads to a lower FID score. Moreover, we see that all the FID scores are also consistent with the qualitative assessment made in Figures 2 and 3 (updated version of the paper).
>
> | Model | Timesteps | FID Score |
> |---------|-------------|-------------|
> | DRFM | 100 | **453.87** |
> | U-Net | 100 | 463.28 |
> | NN | 100 | 457.21 |
> | RF | 100 | 470.12 |
>
> Table: FID scores for generative task
>
> | Model | Timesteps | FID Score |
> |---------|-------------|-------------|
> | DRFM | 100 | 394.99 |
> | U-Net | 100 | 388.38 |
> | NN | 100 | **378.18** |
> | RF | 100 | 450.23 |
>
> Table: FID scores for denoising task
>
> *2. The paper currently compares the DRFM solely with a fully-connected layer. Given the prominence of both the Unet and transformer architectures in noise prediction networks, it would be beneficial for the authors to discuss the relationship of DRFM to these models. In particular, considering that transformer layers utilize a combination of the attention mechanism and fully-connected MLPs, they align more closely with the scope of the fully-connected layer baseline. It would be valuable to explore this connection further.*
>
> **Answer:**
>
> - As per the suggestion, we have now added a few more results using a simple U-Net architecture with the number of trainable parameters similar to the DRFM model. As we see in Figure 2 in the revised version of the paper, for generative tasks, the U-Net performs very similar to that of DRFM.
>
> *3. Since the authors have generalized the random features with the properties of score-matching, it remains unknown whether DRFM can improve the performance in the tasks evaluated in Kawaguchi et al. (2018).*
>
> **Answer:**
>
> - The tasks in Kawaguchi et al. (2018) are different from the goal of this paper. Our goal is to particularly develop an interpretable model that can be potentially used for understanding diffusion models. Moreover, the model architecture in Kawaguchi et al. (2018) does not involve a trainable time parameter which makes our proposed model much different.

---

> > ### Comment · Reviewer_crxL · 2023-11-23
> >
> > I appreciate the authors' response. After reading through the revision and provided additional results, a part of my concerns are gone. Although I believe the method and error quantification is important, some key points are not sufficiently elaborated in the paper, e.g., the paper is a little incremental compared to Chen et. al. (2022), how to handle the efficiency of the model (as we need to increase the number of feature N >> Kd) is unclear. Unfortunately, given these concerns, I regret to consider the manuscript is not yet ready to publish.

---

### Official Review · Reviewer_2YaV · 2023-10-30

**Soundness:** 1 poor
**Presentation:** 3 good
**Contribution:** 1 poor
**Rating:** 3
**Confidence:** 5

**Summary:**

This paper proposes to model the score of image distributions with a one-hidden-layer random features network. The first layer weights are frozen to their initialization, while the second layer weights are learned and depend on the noise level (through a factorization to reduce the number of parameters). The model is compared with a classical one-hidden-layer network where the first layer is learned and a random features model where the dependence on the noise level is not trained.

**Strengths:**

The paper is clearly written (except for section 2 which redundantly exposes both the discrete and continuous formulations of diffusion models).
The parameterization of the second-layer weights dependence on the noise level is interesting and to the best of my knowledge novel.

**Weaknesses:**

As noted in the final sentence of the conclusion, there is nothing deep about this random features model (despite the name), leading to a very weak expressivity that is plagued by the curse of dimensionality. The paper thus studies a very restricted, toyish setting. This is not a problem for a theoretical paper, but the theoretical analysis is lacking in rigour and novelty. In particular, Theorem 3.3 is a straightforward combination of classical results (from Rahimi and Recht and Chen et al.) and does not state key assumptions (namely, that the true scores of the image distribution are in some function class for all noise levels). It also only tackles score approximation, not optimization and generalization from a finite training set, but this is not stated in the text. Finally, the bound holds in probability, but is stated as if it were almost sure. On the numerical side, there are no quantitative comparisons between the different approaches, and I'm frankly not convinced that the proposed approach really outperforms the one-hidden-layer neural network.

**Questions:**

Minor suggestions:
- choose either discrete or continuous framework to present diffusion models
- Typo on page 4: "features [if -> of] the input data"
- page 5: "random and trainable" is confusing phrasing
- notation $p_\theta$ for the network is confusing (usually used for probability density), prefer e.g. $\varepsilon_\theta$
- Put definition of $\mathcal F_\omega$ in the main text if it is used in a lemma
- How can denoising requires 100 steps when it should just be a direct evaluation of the score network? This should be explained in the text.

---

> ### Author Response · Authors · 2023-11-22
> **Response to Reviewer 2YaV (Part 1)**
>
> Thank you for reviewing our manuscript. We appreciate the constructive feedback. We have carefully updated the manuscript to address your concerns. We have marked the changes in the main text in blue. Our response to the concerns are given below.
>
> *On the numerical side, there are no quantitative comparisons between the different approaches,} and I'm frankly not convinced that the proposed approach really outperforms the one-hidden-layer neural network.*
>
> Answer: As per the suggestion, we have now added a few more results using a simple U-Net architecture with the number of trainable parameters similar to the DRFM model. As we see in Figure 2, for generative tasks, the U-Net performs very similar to that of DRFM. We have also added Table 3 that gives the Fr\'echet Inception Distance (FID) calculated using a batch of 50 images from the training dataset and 15 of the generated images by each of the models. Note that the scores values given are for the sake of comparing the four methods and more samples may improve the FID score. We see that for the generative task, the the proposed DRFM architecture gives the lowest scores. The more commonly used U-Net model is at the third position after the NN model. For the denoising task, we see that the NN gives the best results, (although some of the images are inconsistently noisy), followed by U-Net and then DRFM. Note that for U-Net, while most images are denoised perfectly, some images are incompletly formed which leads to a lower FID score. Moreover, we see that all the FID scores are also consistent with the qualitative assessment made in Figures 2 and 3.
>
> - Table with FID scores for generative task
> | Model | Timesteps | FID Score|
> | ----------- | ----------- |---------|
> | DRFM | 100 | **453.87**|
> |U-Net | 100 | 463.28|
> | NN | 100 | 457.21|
> | RF | 100 | 470.12|
>
> - Table with FID scores for denoising task
> | Model | Timesteps | FID Score|
> | ----------- | ----------- |---------|
> | DRFM | 100 | 394.99|
> |U-Net | 100 | 388.38|
> | NN | 100 | **378.18**|
> | RF | 100 |450.23|
>
> Thank you for providing constructive suggestions.
> -*choose either discrete or continuous framework to present diffusion models*
>
> Answer: Our initial purpose for including the continous SDE is to recall the use the contious framework to show the connection between the loss function used to predict noise (Eq.8 in our paper) and the score matching formula (Eq. 13 in our paper). Apparently, there is a direct connection without using the continuos framework. Our training model is based on the discrete framework.
>
> -*Typo on page 4: "features [if -> of] the input data"*
>
> Answer: Fixed.
>
> *page 5: "random and trainable" is confusing phrasing*
>
> Answer: We remove "random" in the phrase for clarity.
>
> *Put definition of $\mathcal{F}_\omega$ in the main text if it is used in a lemma
>
> Answer: Done.

---

> > ### Comment · Reviewer_2YaV · 2023-11-23
> >
> > Thank you for your answer. I must say that FID values of ~400 are way out of the reasonable range to consider any of the compared networks from having learned a good model of the data distribution (also, 50 images is simply not enough to correctly estimate a FID score). I therefore keep my score and recommendation.

---

> > > ### Author Response · Authors · 2023-11-23
> > >
> > > As mentioned in the revised version of the paper, the FID scores in the paper are not meant to represent the true distribution as we currently calculate it over a very small dataset simply for the sake of have a quantitative metric for comparing different methods.
> > >
> > > Thank you for your response and the feedback.

---

### Official Review · Reviewer_3pnu · 2023-11-03

**Soundness:** 3 good
**Presentation:** 4 excellent
**Contribution:** 3 good
**Rating:** 3
**Confidence:** 4

**Summary:**

In this work, the authors propose a diffusion model-inspired random feature model. This model is interpretable and is able to achieve comparable numerical results to a fully connected neural network (of the same size). The authors then give generalization bounds between the distribution of sampled data and the true distribution for their proposed model.

**Strengths:**

- The authors propose a diffusion RF model. The model they propose is quite simple and interpretable.
- The model achieves the same performance as a fully connected network on some toy datasets.
- The generalization bound in Theorem 3.3 is novel and interesting.

**Weaknesses:**

- Most of the heavy lifting in the proof of Theorem 3.3 is done in Lemma A.2 which is from Chen et al. (2022).

- The authors only experiment with very simple and small datasets such as fashion MNIST. It is not that clear what happens for more complex datasets. Here, in the diffusion RF model, feature learning is absent because W is fixed at initialization. It is hard to believe that the model can be successful for more complex tasks.

- The main goal of the paper is not clear to me. Are the authors proposing the RF model as a tool to decrease computational complexity (like Rahimi and Recht) or some theoretical tool to analyze a more sophisticated phenomenon (like Mei and Montanari, 2019)? Do the authors expect such models to achieve similar accuracies to state of the art methods?

I agree that there models are analyzable; e.g., Theorem 3.3 is interesting. However, what phenomenon is this model tryin to describe? For example (Mei and Montanari, 2019) and others use RF models to analyze double descent and to show that this phenomenon exists even in very simple models.

- What motivates this particular choice of random features? Why did you choose W to be the random weights and train the other parameters?

- I think in the model that the authors are analyzing, it is more interesting to study the case where the dimension d is proportional to N. Can the authors explain what will happen in that regime to the bound in (19)? In general, I think a more thorough discussion after Theorem 3.3. would benefit the paper a lot. What do we understand from the generalization bound?

**Questions:**

Please see above.

---

> ### Author Response · Authors · 2023-11-22
>
> Thank you for reviewing our manuscript. We appreciate the constructive feedback. We have carefully updated the manuscript to address your concerns. We have marked the changes in the main text in blue. Our response to the concerns are given below.

---

> ### Author Response · Authors · 2023-11-22
> **Response to comments on weakness**
>
> *1. Most of the heavy lifting in the proof of Theorem 3.3 is done in Lemma A.2 which is from Chen et al. (2022). The authors only experiment with very simple and small datasets such as fashion MNIST. It is not that clear what happens for more complex datasets. Here, in the diffusion RF model, feature learning is absent because W is fixed at initialization. It is hard to believe that the model can be successful for more complex tasks.*
>
> **Answer:**
>  - The fixing of weights and biases in first layer is a common approach in random feature models. The idea is based on using large enough weights and biases (fixed) such that they can approximate shift invariant kernels, see Rahimi and Recht (2007) for more details. For example, Fourier-type feature functions represent the Gaussian kernel when the components of matrix $\mathbf{W}$ are sampled from Gaussian distribution and bias vector $\mathbf{b}$ is sampled from the uniform distribution. Since the aim of diffusion models is to learn the mean (and covariance) of the reverse process whose transition kernels are Gaussian distributions, we choose this particular form. For more details on the relation between Fourier-type features and Gaussian kernel, we refer to Rahimi and Recht (2007).
>  - In addition to the fashion MNIST dataset, we also provide some experiments based on music samples to test our method for different types of datasets. Additionally, we have added experiments using U-Net, a more commonly used architecture for diffusion models (Figure 2 and 3 in revised version) along with a table of FID scores (Table 1 and also given below).
> - Table with FID scores for generative task
> | Model | Timesteps | FID Score|
> | ----------- | ----------- |---------|
> | DRFM | 100 | **453.87**|
> |U-Net | 100 | 463.28|
> | NN | 100 | 457.21|
> | RF | 100 | 470.12|
>
> - Table with FID scores for denoising task
> | Model | Timesteps | FID Score|
> | ----------- | ----------- |---------|
> | DRFM | 100 | 394.99|
> |U-Net | 100 | 388.38|
> | NN | 100 | **378.18**|
> | RF | 100 |450.23|
>
> - We see that for the generative task, the the proposed DRFM architecture gives the lowest scores. The more commonly used U-Net model is at the third position after the NN model. For the denoising task, we see that the NN gives the best results, (although some of the images are inconsistently noisy), followed by U-Net and then DRFM. Note that for U-Net, while most images are denoised perfectly, some images are incompletely formed which leads to a lower FID score. To summarize, we see that all the three methods i.e., DRFM, NN and U-Net are comparable although the last two do not enjoy the theoretical interpretability that DRFM does.
>
> *2. The main goal of the paper is not clear to me. Are the authors proposing the RF model as a tool to decrease computational complexity (like Rahimi and Recht) or some theoretical tool to analyze a more sophisticated phenomenon (like Mei and Montanari, 2019)? Do the authors expect such models to achieve similar accuracies to state of the art methods?*
>
> **Answer:**
> - **Goal:** The goal of this paper is to propose a model that can be used as a theoretical tool to understand the training process of diffusion models and have a quantified method to bound the sampling error based on the properties of the model. For example, reformulation of DRFM into random features shows that there is one random feature model corresponding to each timestep. Furthermore, by choosing $N\geq Kd$ (as we chose in the experiments as well), we can bound the sampling error $TV(p_{\theta}(0),q)\leq\mathcal{O}(\varepsilon)$ which would otherwise be $TV(p_{\theta}(0),q)\leq\mathcal{O}(\varepsilon+\varepsilon_{\text{score}})$.
>
>  - For this current work, we have compared to some similar methods and also added some additional experiments with more commonly used architectures such as the U-Net in the revised version of the paper. In the current setting, we do not see a significant difference between the performance of DRFM and the other models, although none of the other models such as NN and U-Net enjoy the interpretability as DRFM. We leave further exploration using more complex models and tasks for future work.

---

> ### Author Response · Authors · 2023-11-22
> **Response to further questions by Reviewer 3pnu**
>
> *1. Theorem 3.3 is interesting. However, what phenomenon is this model tryin to describe? For example (Mei and Montanari, 2019) and others use RF models to analyze double descent and to show that this phenomenon exists even in very simple models.*
>
> **Answer:**
>
> In Theorem 3.3 we quantify the upper bounds for the total variation between the true distribution and learned distribution. In particular, we show that when DRFM is used for learning the distribution, we can choose the number of features $N$ in such a way that the error due to the parameterized model becomes small.
>
> *2. What motivates this particular choice of random features? Why did you choose W to be the random weights and train the other parameters?*
>
> **Answer:**
>  - We focus mainly on Fourier-type feature functions as they represent the Gaussian kernel when the components of matrix $\mathbf{W}$ are sampled from Gaussian distribution and bias vector $\mathbf{b}$ is sampled from the uniform distribution. Since the aim of diffusion models is to learn the mean (and covariance) of the reverse process whose transition kernels are Gaussian distributions, we choose this particular form. For more details on the relation between Fourier-type features and Gaussian kernel, we refer to Rahimi and Recht (2007).
>  - The idea of fixing first layer weights and training the others is based on the training of random feature models. The feature function with fixed weights is a way to approximate a kernel depending on the distribution from which the weights are sampled. Detailed explanation and derivations can be found in Rahimi and Recht (2007, 2008) and Rudi and Rosasco (2017).
>
> *3. I think in the model that the authors are analyzing, it is more interesting to study the case where the dimension d is proportional to N. Can the authors explain what will happen in that regime to the bound in (19)? In general, I think a more thorough discussion after Theorem 3.3. would benefit the paper a lot. What do we understand from the generalization bound?*
>
> **Answer:**
>
>  - Note that the results of random features in general hold in the overparameterized setting. Unless some additional information is known about the target function or the underlying distribution, the number of features $N$ can not be reduced arbitrarily. For example, for a simple function approximation problem, if the target function is a sparse additive function, then the number of features $N$ can be reduced as explained in Hashemi et. al. (2023), Rudi and Rosasco (2017).
>
> - The quantification in Theorem 3.3 helps us to choose the appropriate number of model parameters before training. In particular, note that for appropriate choices of $T$ and $h$, the bounds in Chen et. al. (2022) can be atmost reduced to $TV(p_{\theta}(0),q)\leq\mathcal{O}(\varepsilon+\varepsilon_{\text{score}})$. Our results show that we can also choose the parameters $N\gg Kd$ in the model so that $TV(p_{\theta}(0),q)\leq\mathcal{O}(\varepsilon)$. We have briefly pointed this out in Section 1 of our revised version.

---

> ### Comment · Reviewer_3pnu · 2023-11-22
>
> I thank the authors for the response. However, I am still not satisfied by the general motivation behind such model/analysis and will keep my initial score.

---

### Official Review · Reviewer_2JVx · 2023-11-03

**Soundness:** 3 good
**Presentation:** 3 good
**Contribution:** 3 good
**Rating:** 6
**Confidence:** 3

**Summary:**

The paper proposes to use random features in the training of diffusion models for interpretability and ease of computation. In more details, the authors estimate the score function by a class of functions generated by random features. They demonstrate the method on FASHION-MNIST and Audio data.

**Strengths:**

The proposed method replaces deep neural nets, which are commonly used when approximating score functions, with random feature functions and shows that experimentally the new method is competitive.

**Weaknesses:**

1. The theorem 3.1 essentially combines error decomposition results from diffusion models and approximation results from random feature functions. Is it possible to have more fine-grained results for the specific feature map (Eq 17) chosen in the paper?

2. How do other choices of random feature maps behave compared to the sin/cos map chosen in the current paper? A sensitivity analysis would improve the validity of the paper.

**Questions:**

See weaknesses.

---

> ### Author Response · Authors · 2023-11-22
> **Response to comments on weakness**
>
> **Thank you for reviewing our manuscript. We appreciate the constructive feedback. We have summarized our responses to address your concerns below.**
>
> *1. The theorem 3.1 essentially combines error decomposition results from diffusion models and approximation results from random feature functions. Is it possible to have more fine-grained results for the specific feature map (Eq 17) chosen in the paper?*
>
> **Answer:**
>
> - Equation (15) does not look like a classical random feature model explicitly. However, when we expand the formula, we see that equation (15) can be written as a random feature model. Theoretically, we show that the class of functions approximated by our proposed model can be rewritten as in Eq. (16) and (17) which we prove to be the same as that of random features (see proof of Lemma 3.1). Thus a separate analysis would yield the same results.
> - Although the learning problem is nonconvex, DRFM has no bad local minimum.
>  - Without the reformulation of DRFM as a random feature model, having a theoretical analysis is as challenging as attempting to analyze a neural network.
>  - However, when training, the form in equation (15) plays an important role as it involves weights of the time parameter without which the numerical results would fail (as demonstrated by the numerical results using RF method).
>
>
> *2. How do other choices of random feature maps behave compared to the sin/cos map chosen in the current paper? A sensitivity analysis would improve the validity of the paper.*
>
> **Answer:**
>
> - Theoretically, any activation function (instead of sin/cos) would suffice as long as the activation function is bounded as indicated in Rahimi and Recht (2007).
> - While numerical results might differe slightly based on the choice of feature functions, we focus mainly on Fourier-type feature ($\sin(.)$ for the feature type) functions as they represent the Gaussian kernel when the components of matrix $\mathbf{W}$ are sampled from Gaussian distribution and bias vector $\mathbf{b}$ is sampled from the uniform distribution. Since the aim of diffusion models is to learn the mean (and covariance) of the reverse process whose transition kernels are Gaussian distributions, we choose this particular form. For more details on the relation between Fourier-type features and Gaussian kernel, we refer to Rahimi and Recht (2007).
> - Our choice of cosine function for representing the time parameter reflects the idea of positional encoding as commonly used in classical diffusion models. However, theoretically as mentioned above, any bounded function would suffice.

---

### Official Review · Reviewer_vFu7 · 2023-11-06

**Soundness:** 1 poor
**Presentation:** 3 good
**Contribution:** 1 poor
**Rating:** 3
**Confidence:** 4

**Summary:**

The paper applies the idea of random features model to the diffusion-based generative model. Specifically, the paper proposes using a denoiser of the form Eq. (15), where $W, b$ are fixed and $\theta^1, \theta^2$ are trained. Then, the paper presents theoretical results regarding the TV distance between the true distribution and the estimated distribution. Experiments on fashion-MNIST and audio data are provided, and the advantage over other methods is discussed.

**Strengths:**

- The paper provides a well-written description of diffusion model and related concepts.

- The proposed algorithm is straightforward to understand.

- Combining random features model and diffusion model is a novel approach.

**Weaknesses:**

- The random features model was first introduced as an approximation scheme for the kernel method and later used as a toy model for studying neural networks. I personally understand it as a theoretical tool rather than a practical algorithm. Therefore, I think that the idea of using the random features model in generative modeling is not particularly interesting unless it provides a fundamentally new understanding of diffusion-based generative modeling. While the paper provides some theoretical results, they appear to be simple extensions of previously known results.

- To my understanding, the finding of the experiment section can be summarized as "training only $\theta^1, \theta^2$ showed better performance than training all parameters or training only $\theta^2$". Can the authors provide results for different $N$'s? The current experiment is using huge $N$, and it is not surprising that restricting the model class can lead to a better generalization.

**Questions:**

- Can the authors provide other popular evaluation metrics for generative models such as inception score, Frechet inception distance, etc.?

---

> ### Author Response · Authors · 2023-11-22
> **Authors' response to official review of submission5657 by reviewer vFu7**
>
> *1. I think that the idea of using the random features model in generative modeling is not particularly interesting unless it provides a fundamentally new understanding of diffusion-based generative modeling. While the paper provides some theoretical results, they appear to be simple extensions of previously known results.*
>
> **Answer:**
>
> - Using random features for diffusion models could be one of the ways in which diffusion models can be understood. Note that random features are an approximation of (shift invariant) kernels depending on the choice of distribution of the fixed weights. Since transition kernels are essentially Gaussian distributions in diffusion models, our proposed method is the first step to exploring diffusion models with interpretable architectures.
>
> - The performance of random feature models can be considered superior to other similar surrogate models such as neural networks. They have a function space similar to that of shallow neural networks (the function class of random feature models is dense in the space of two-layer neural networks) and are computationally cheaper and easier to implement. In comparison to deeper neural networks which are a black-box, random feature models can quantify the generalization bound based on the number of features and measurements required.
>
> - Although the results may seem extensions of existing works, we would like to point out that the proposed architecture is not explicitly a random feature model. Firstly, it involves two layers of training. Secondly, it involves a certain involvement of time parameter which is uncommon to the classical random feature or the semi-random feature approach. Our major contribution lies in reducing the proposed architecture to a random feature-based model and applying the extended approximation bound (Lemma 3.2) to conclude that the bounds in Chen et. al. (2022) can be reduced from $TV(p_{\theta}(0),q)\leq\mathcal{O}(\varepsilon+\varepsilon_{\text{score}})$ to $TV(p_{\theta}(0),q)\leq\mathcal{O}(\varepsilon)$ with an appropriate choice of the number of parameters.
>
> *2. To my understanding, the finding of the experiment section can be summarized as "training only
>  $\theta^1,\theta^2$ showed better performance than training all parameters or training only $\theta^2$. Can the authors provide results for different
> N's? The current experiment is using huge
> N, and it is not surprising that restricting the model class can lead to a better generalization.*
>
> **Answer:**
>
> - From the theoretical results presented in the paper, we need to choose $N\gg Kd$ to achieve a small error for our proposed method. Thus for our experiments we choose $N=80000$ ($N\gg Kd$, where $Kd = 100\times 28\times 28$). From previous works on random features in Rahimi and Recht (2007, 2008), Rudi and Rosasco (2017), the results for random feature models hold in an overparameterized setting.
>
> *3. Can the authors provide other popular evaluation metrics for generative models such as inception score, Frechet inception distance, etc.?*
>
> **Answer:**
>
> Thank you for the suggestion. We have now included the FID scores comparing different methods in the revised version of the paper.
>
> - Table 1 in the updated version of the paper (also given below) gives the Fr\'echet Inception Distance (FID) calculated using a batch of 50 images from the training dataset and 15 of the generated images by each of the models. Note that the scores values given are for the sake of comparing the four methods and more samples may improve the FID score. We see that for the generative task, the the proposed DRFM architecture gives the lowest scores. The more commonly used U-Net model is at the third position after the NN model. For the denoising task, we see that the NN gives the best results, (although some of the images are inconsistently noisy), followed by U-Net and then DRFM. Note that for U-Net, while most images are denoised perfectly, some images are incompletely formed which leads to a lower FID score. Moreover, we see that all the FID scores are also consistent with the qualitative assessment made in Figures 2 and 3 (see updated version of the paper).
>
> - Table with FID scores for generative task
> | Model | Timesteps | FID Score|
> | ----------- | ----------- |---------|
> | DRFM | 100 | 453.87|
> |U-Net | 100 | 463.28|
> | NN | 100 | 457.21|
> | RF | 100 | 470.12|
>
> - Table with FID scores for denoising task
> | Model | Timesteps | FID Score|
> | ----------- | ----------- |---------|
> | DRFM | 100 | 394.99|
> |U-Net | 100 | 388.38|
> | NN | 100 | 378.181|
> | RF | 100 |450.23|

---

> > ### Comment · Reviewer_vFu7 · 2023-11-22
> >
> > Thanks for the response. As I am still not convinced by the main contribution of the paper., I will keep my score.

---

### Official Review · Reviewer_58mK · 2023-11-09

**Soundness:** 3 good
**Presentation:** 2 fair
**Contribution:** 3 good
**Rating:** 5
**Confidence:** 3

**Summary:**

This paper proposes a new model, called a diffusion random feature model (DRFM), that incorporates the interpretability of random feature models with the data-generating capability of diffusion models. This hybrid model, DRFM, aims to produce results comparable to fully connected neural networks while maintaining interpretability and using a comparable number of trainable parameters. The model's effectiveness is tested through experiments on the fashion MNIST dataset and instrumental audio data. The results demonstrate that DRFM can learn to generate data from a limited number of training samples (as few as one hundred) and within a limited number of timesteps (one hundred).

**Strengths:**

The paper highlights the interpretability and computation efficiency of the DRFM, noting that it allows for the derivation of theoretical upper bounds on the quality of the samples generated. Numerical experiments show that DRFM outperforms both a traditional fully connected network with all layers trainable and a standard random features model where only the last layer is trainable. The result is clear and opens up a theoretical direction for analyzing random feature models with high-dimensional spaces for diffusion models.

**Weaknesses:**

1. I did not get the point of focusing on random Fourier-type feature models in Eq. (15). Following, the proof of the paper, it is possible to include more general activation functions. I did not find the motivation why we only consider this specific random feature model defined in Eq. (15) and Eq. (16). The authors may need to provide additional explanation of the choice of the random feature or empirical evidence of the advantage of this specific random feature model.

2. The theory of this paper seems to directly come from Chen et al. (2022) and Ranhimi & Recht (2008b; a). The authors may need to emphasize the difference between the current paper and the references, and the difficulty in the analysis in the current paper. In particular, the authors derived the generalization bounds via the approximation error of the random feature model, which, I think, lacks explanations in the proof in the Appendix.

3. Although there may not be a theory, for completeness, the authors may need to provide additional simulations for deeper neural networks with random features, showing the advantage of this architecture, and comparing it with more commonly used U-Net or other neural network models. The random feature models will reduce the computation complexity but I am not sure the performance is still comparable with conventional diffusion models.

**Questions:**

1. It is a standard practice to introduce an abbreviation by providing its full form for the first time. For instance, DDPM appears in the paper first time without explaining the full name.

2. Between (5) and (6), there are typos in $q(x_{k-1}|x_k,x_0)=\mathcal{N}(\tilde{\mu}_t,\tilde{\beta}_t)$: you should specify $t$ and insert identity matrix for covariance. And the same issue for (6).

3. Please explain ''U-Nets not only preserve the dimension of input data, they also apply techniques such as downsampling using
convolution which helps to learn the features if the input data.'' on page 4.

4. In Algorithm 1, line 4, did you use uniformly sampling from $K$ time points to update the training parameters? For line 7, how do you minimize the loss $L$? Did you assume it will attain some global minimizer after training? Further explanations may be needed.

5. In Lemma 3.1, please explain the notion $\mathcal{F}_{\omega}$ and $\mathbf{\theta}_j^{(2)}$.

6. In Lemma 3.2, is $\rho(\omega)$ the density function of $\rho$?

7. In Theorem 3.3, you assume a bounded second moment for $q(x_0)$ but Chen et al. (2022) require $(2+\eta)$-th moment bound for some $\eta>0$. Is there any extra work to relax this assumption in your proof? Right now, there is a lack of explanation of the proof and how you applied Lemmas 3.1, 3.2, and A.2 to conclude the main theorem. Lemma 3.2 shows the approximation error of the random feature model but in the DRFM, how do you use (18) to finish the proof of Theorem 3.3 and ensure $|\theta^{(2)}_{ij}|$ are always uniformly bounded by some constant?

8. Any explanation for why neural networks may fail in denoising images in Figure 3 sometimes but DRFM works better and is stable?

9. There should be more details of how you trained the models in the captions of Figures 2-7, e.g. which optimizers are used.

---

> ### Author Response · Authors · 2023-11-21
> **Responses to Comments on Weaknesses**
>
> *1. I did not get the point of focusing on random Fourier-type feature models in Eq. (15). Following, the proof of the paper, it is possible to include more general activation functions. I did not find the motivation why we only consider this specific random feature model defined in Eq. (15) and Eq. (16). The authors may need to provide additional explanation of the choice of the random feature or empirical evidence of the advantage of this specific random feature model.*
>
> Answer:
> 1. In general, any activation function would suffice as long as the activation function is bounded as indicated in Rahimi and Recht (2007).
>
> 2. We focus mainly on Fourier-type feature functions as they represent the Gaussian kernel when the components of matrix $\mathbf{W}$ are sampled from Gaussian distribution and bias vector $\mathbf{b}$ is sampled from the uniform distribution. Since the aim of diffusion models is to learn the mean (and covariance) of the reverse process whose transition kernels are Gaussian distributions, we choose this particular form. For more details on the relation between Fourier-type features and Gaussian kernel, we refer to Rahimi and Recht (2007).
>
>
> *2. The theory of this paper seems to directly come from Chen et al. (2022) and Ranhimi & Recht (2008b; a). The authors may need to emphasize the difference between the current paper and the references, and the difficulty in the analysis in the current paper. In particular, the authors derived the generalization bounds via the approximation error of the random feature model, which, I think, lacks explanations in the proof in the Appendix.*
>
> Answer: Thank you for the suggestions. We have added the following discussion to the references in the Introduction section.
>
> - Firstly, we show that our proposed architecture is a random feature model which helps us to obtain the results given in the paper. Also, we would like to highlight the fact that our model architecture is capable of incorporating time as an input with a separate set of weights associated with it. This is much different from the architecture proposed in Kawaguchi et al. (2018) where the authors only consider the input $\mathbf{x}$.
>
> - Secondly, we would like to point out that although our final result is based on results from Chen et. al. (2022), we believe that the quantification of $\varepsilon_{\text{score}}$ is a significant contribution to understanding the number of parameters needed (with respect to the input dimension and number of timesteps) to obtain a small error. This quantification helps us to choose the appropriate number of model parameters before training. In particular, note that for appropriate choices of $T$ and $h$, the bounds in Chen et al. (2022) can be atmost reduced to $TV(p_{\theta}(0),q)\leq\mathcal{O}(\varepsilon+\varepsilon_{\text{score}})$. Our results show that we can also choose the parameters of the model so that $TV(p_{\theta}(0),q)\leq\mathcal{O}(\varepsilon)$.
>
> - Also, we explicitly extend existing function approximation bounds to hold for $d-$ dimensional function. Although trivial, this result forms the basis for the main result as well which has not been used in any previous works.
>
> - Lastly, as for the proof in the appendix, we try to be as brief as possible to avoid repetition of steps which can be found in the references. For example, the main steps for proving Theorem 3.3 can be found in Chen et al. (2022) and thus we omit the details. However, we provide statements and proofs of all the results corresponding to our proposed method (Equation (16) in Section 3.1, Lemma 3.1 and Lemma 3.2) which are key for the result in Theorem to hold.
>
> *3. Although there may not be a theory, for completeness, the authors may need to provide additional simulations for deeper neural networks with random features, showing the advantage of this architecture, and comparing it with more commonly used U-Net or other neural network models. The random feature models will reduce the computation complexity but I am not sure the performance is still comparable with conventional diffusion models.*
>
> Answer: We have now added a few more results using a simple U-Net architecture with the number of trainable parameters similar to the DRFM model. For the generative task, the performance of DRFM and U-Net are comparable (see Figure 2 in the updated file). For the denoising task, both U-Net and NN perform marginally better, but neither of them has the interpretable properties of DRFM.

---

> ### Author Response · Authors · 2023-11-22
> **Responses to Questions from Reviewer 58mK**
>
> *1. It is a standard practice to introduce an abbreviation by providing its full form for the first time. For instance, DDPM appears in the paper first time without explaining the full name.*
>
> Answer: Thank you for pointing out this typo. It is fixed.
>
> *2. Between (5) and (6), there are typos in ...*
>
> Answer: Thank you for pointing out this typo. It is fixed.
>
> *3. Please explain ''U-Nets not only preserve the dimension of input data, they also apply techniques such as downsampling using convolution which helps to learn the features if the input data.'' on page 4.*
>
> Answer: The architecture of U-Net is designed in a way that it preserves the input dimension of the data i.e., the output and input dimension are the same. Additionally, U-Nets generally consist of a contracting (downsampling) and expanding (upsampling) path. The contracting path consists of sequences of convolutional and max-pooling layers that downsample the input images and extracts the layers.
>
> *4. In Algorithm 1, line 4, did you use uniformly sampling from $K$ time points to update the training parameters? For line 7, how do you minimize the loss $L$? Did you assume it will attain some global minimizer after training? Further explanations may be needed.*
>
> Answer:
> - We used uniform sampling from 100 timesteps to update our parameters. This is consistent with the algorithm outlined in DDPM.
> -We minimize the mean-squared error loss to update the weights using the ADAM optimizer.
> -Note that training using random features is a convex method and thus we expect our proposed model to reach a global minimizer post-training.
>
> *5. In Lemma 3.1, please explain the notion $\mathcal{F}_{\boldsymbol{\omega}}$ and $\boldsymbol{\theta}_j^{(2)}$. In Lemma 3.2, is $\rho(\boldsymbol{\omega})$ the density function of $\rho$?*
>
> Answer: We have updated the paper by rearranging some of the definitions which was initially given later. We denote $\boldsymbol{\theta}_j^{(2)}$ the $j^{th}$ row of the matrix $\boldsymbol{\theta}^{(2)}\in\mathbb{R}^{N\times d},\rho(\boldsymbol{\omega})$ the probability density function from which $\boldsymbol{\omega}$ is sampled, and
>
> $\mathcal{F}_{\boldsymbol{\omega}}$ be the set of all functions $\boldsymbol{f}(\mathbf{x})$ of the form:
>
> $\sum\limits_{j=1}^N \boldsymbol{\alpha_j} \,\phi(\mathbf{x}^T\boldsymbol{\omega}_j)$
>
> such that $||\boldsymbol{\alpha_j}||_{\infty}\leq \frac{C}{N}$ for all $j=1,\ldots N$.
>
> *7. In Theorem 3.3, you assume a bounded second moment for but Chen et al. (2022) require $(2+\eta)$-th moment bound for some. Is there any extra work to relax this assumption in your proof?} Right now, there is a \textcolor{blue}{lack of explanation of the proof and how you applied Lemmas 3.1, 3.2, and A.2 to conclude the main theorem.} Lemma 3.2 shows the approximation error of the random feature model but in the DRFM, how do you use (18) to finish the proof of Theorem 3.3 and ensure $|\boldsymbol{\theta}_{ij}^{(2)}|$ are always uniformly bounded by some constant?*
>
> Answer:
>
> -In the paper the authors specify, "For technical reasons, we need to assume that $q$ has a finite moment of order slightly but strictly bigger than 2, but our quantitative bounds will only depend on the second moment." Also, we used the proof from a different version of the paper which uses the second moment for the proof (as also mentioned by the authors). However, for consistent results, we have updated the paper to reflect the changes.
>
> -Also, in Lemma 3.1 we show that the class of functions approximated by our proposed method is the same as the space of functions approximated by a random feature model. Further Lemma 3.2 gives the approximation bound for a $d-$dimensional function approximated by random features which will be the same for DRFM. The upper bound leads to the satisfaction of the third assumption of Lemma A.2 which leads to the stated conclusion in Theorem 3.3.
> Any explanation for why neural networks may fail in denoising images in Figure 3 sometimes but DRFM works better and is stable?
>
> *8. Any explanation for why neural networks may fail in denoising images in Figure 3 sometimes but DRFM works better and is stable?*
>
> Answer: Training neural networks is a non-convex problem while training DRFM is a convex problem so DRFM is more stable.
>
> *9. There should be more details of how you trained the models in the captions of Figures 2-7, e.g. which optimizers are used.*
>
> Answer: Thank you for the suggestion. We have added more details on the training process both in the main text of the paper and captions of figures.

---

> ### Author Response · Authors · 2023-11-22
> **Responses to Reviewer 58mK**
>
> Thank you for reviewing our manuscript. We appreciate the constructive feedback.  We have carefully updated the manuscript to address your concerns. We have marked the changes in the main text in blue.

---

### Meta-Review · Area_Chair_3G8A · 2023-12-05

**Metareview:**

The paper proposes combining random feature models with diffusion models. With this, the authors study the theoretical upper bounds on the quality of the samples generated. The authors are one of the first ones to use random feature models to analyze diffusion. I appreciate the authors for attempting to study this problem, as this can potentially open up several directions for analyzing and understanding diffusion models.

However, the paper is not in a state for acceptance at this stage. Reviewers mainly complain that simply using random features model in generative modeling is not particularly interesting unless it provides a fundamentally new understanding of diffusion-based generative modeling, which the paper doesn't seem to. Most of the theoretical contributions are simple extensions of Chen et al. (2022) and Ranhimi & Recht (2008b; a). Since the theoretical contributions are weak, the reviewers feel that the contributions are not sufficient for publishing at ICLR.

I would encourage authors to pursue this direction further and study this problem in detail, and resubmit in a future venue.

**Justification For Why Not Higher Score:**

Theoretical contributions are weak. Results are simple extensions of prior approaches.

**Justification For Why Not Lower Score:**

N/A

---

### Decision · Program_Chairs · 2024-01-16

Reject